# UHRF2 mediates resistance to DNA methylation reprogramming in primordial germ cells

Ambre Bender [1,2], Marion Morel[1,2], Michael Dumas [1,2], Muriel Klopfenstein[3], Naël Osmani[1,4,5,6], Maxim V. C. Greenberg [7,8], Déborah Bourc'his [7], Norbert B. Ghyselinck [3] & Michael Weber [1,2]

In mammals, primordial germ cells (PGCs) undergo global erasure of DNA methylation with delayed demethylation of germline genes and selective retention of DNA methylation at evolutionarily young retrotransposons. However, the molecular mechanisms of persistent DNA methylation in PGCs remain unclear. Here we report that resistance to DNA methylation reprogramming in PGCs requires UHRF2, the paralog of the DNMT1 cofactor UHRF1. PGCs from *Uhrf2* knock-out mice show loss of retrotransposon DNA methylation, while DNA methylation is unaffected in somatic cells. This is not associated with changes in the expression of retrotransposons in E13.5 PGCs, indicating that other mechanisms compensate for retrotransposon control at this stage. Furthermore, *Uhrf2*-deficient PGCs show precocious demethylation of germline genes and overexpress meiotic genes in females. Subsequently, *Uhrf2*-deficient mice show impaired oocyte development and female-specific reduced fertility, as well as incomplete remethylation of retrotransposons during spermatogenesis. These findings reveal a crucial function for the UHRF1 paralog UHRF2 in controlling DNA methylation in the germline.

DNA methylation is an epigenetic mark with crucial functions in vertebrates. The addition of methyl groups occurs on the fifth carbon of cytosines predominantly in the CpG context. The mouse genome encodes four active DNA methyltransferases (DNMT1, DNMT3A, DNMT3B, DNMT3C), as well as the catalytically inactive DNMT3L. DNMT3C and DNMT3L are expressed in germ cells and their inactivation leads to infertility[1,2], whereas the inactivation of DNMT1, DNMT3A, or DNMT3B leads to embryonic or post-natal lethality[3], confirming the essentiality of DNA methylation in mammalian development. In somatic cells, DNA methylation of CG-rich sequences is mainly used to repress imprinted genes, germline genes, and young transposable elements[4].

DNA methylation levels are globally reduced after fertilization and re-established at the time of implantation in mouse embryos[5,6]. A second and more drastic wave of DNA methylation reprogramming occurs in primordial germ cells (PGCs), the embryonic precursors of the gametes. In mice, PGCs arise as a cluster of approximately 40 cells at the base of the incipient allantois around embryonic day 7.25 (E7.25), and then divide and migrate to colonize the gonadal ridge by approximately E11.5[7,8]. During this period, PGCs undergo massive epigenetic reprogramming and erasure of DNA methylation, before DNA methylation is subsequently re-established from E14.5 on in male prospermatogonia and postnatally in female growing oocytes[9]. Several groups investigated the patterns of DNA methylation reprogramming

[1]Université de Strasbourg, Strasbourg, France. [2]CNRS, UMR7242 Biotechnology and Cell Signaling, 300 Bd Sébastien Brant, Illkirch Cedex, France. [3]Institut de Génétique et de Biologie Moléculaire et Cellulaire (IGBMC), Département de Génétique Fonctionnelle et Cancer, CNRS UMR7104, INSERM U1258, Université de Strasbourg, 1 rue Laurent Fries, BP-10142, Illkirch Cedex, France. [4]Tumor Biomechanics, INSERM UMR_S1109, Strasbourg, France. [5]Fédération de Médecine Translationnelle de Strasbourg (FMTS), Strasbourg, France. [6]Equipe Labellisée Ligue Contre le Cancer, Paris, France. [7]Institut Curie, PSL Research University, INSERM U934, Paris, France. [8]Present address: Université Paris Cité, CNRS, Institut Jacques Monod, Paris, France. ✉e-mail: michael.weber@unistra.fr

in PGCs at a limited number of time points[10–13]. These studies revealed that demethylation is initiated in early PGCs before E9.5 and reaches its nadir at E13.5. Mechanisms of demethylation in PGCs involve compromised maintenance of DNA methylation because UHRF1 (ubiquitin-like containing PHD and RING finger domains 1), a cofactor that recruits DNMT1 to replication sites during mitosis, is downregulated and sequestered in the cytoplasm of PGCs[11,14,15]. UHRF1 recognizes hemimethylated DNA and catalyzes ubiquitination of histone H3, which serves as a recruitment signal for DNMT1[16,17]. Global hypomethylation in PGCs also entails conversion of 5mC into 5hmC by TET proteins[18], in particular TET1 which is required for late demethylation of germline and imprinted genes[13,19,20].

Despite a globally hypomethylated landscape, PGCs retain high DNA methylation at a specific subset of young retrotransposons and rare single-copy sequences[10,11,18,21]. The reason why PGCs carry persistent DNA methylation at some retrotransposons is currently unclear. DNA methylation might not be required to maintain repression of retrotransposons because of the compensatory role of SETDB1 and PRC2 during epigenetic reprogramming in PGCs[22,23]. Additionally, although reaching demethylation at E13.5, some sequences undergo a delayed demethylation kinetics during PGC development. Notably, this seems to be essential for halting precocious activation of meiosis-related genes[11,24].

The molecular mechanisms underlying the resistance to DNA methylation erasure in PGCs are unclear. Although PGCs express high levels of DNMT1, they lack the essential activity provided by UHRF1. One hypothesis is that UHRF2, a paralog of UHRF1, could compensate for the absence of UHRF1 to maintain DNA methylation in PGCs at specific loci[25]. UHRF2 shares the same domains as UHRF1, and previous studies demonstrated that UHRF2 interacts with DNMT1, DNMT3A, and DNMT3B[26,27]. Similar to UHRF1, the tandem tudor domain (TTD) and plant homeodomain (PHD) of UHRF2 mediate binding to di- and trimethylation of lysine 9 of histone H3 (H3K9me2/3)[26–28]. UHRF2 is also a reader of hydroxymethylated DNA through its SRA domain[29,30]. UHRF2 seems not implicated in general maintenance methylation because it has no preference for hemimethylated DNA and low histone ubiquitination activity on nucleosome substrates[28]. Indeed, UHRF2 cannot rescue DNA methylation defects in mouse *Uhrf1-/-* ESCs[26,27]. *Uhrf2*-deficient mice were viable and showed discrete defects in DNA methylation and hydroxymethylation in the brain associated with impaired neuronal function[31,32].

Here, we profiled DNA methylation across normal PGC development (from embryonic stages E8.5 to E17.5) and provide quantitative data supporting delayed demethylation of germline genes and selective maintenance of DNA methylation by DNMT1 at retrotransposons (ERVK, ERV1, and L1Md families) during PGC development. We further demonstrate that UHRF2 is required for preventing DNA methylation erasure in PGCs and investigate the consequences of *Uhrf2* deficiency on germ cell transcription, methylation, and development in mice.

## Results
### DNA methylation dynamics in developing PGCs
To complement previous studies that investigated PGC DNA methylation[10–13,21], we first generated a complete methylation atlas of PGC development in the mouse. We used mice carrying an *Oct4(ΔPE)*-GFP transgene[33] to isolate pure populations of PGCs by fluorescence-activated cell sorting (FACS) from E9.5 to E17.5 (Supplementary Fig S1a) and profiled DNA methylation by Reduced Representation Bisulfite Sequencing (RRBS) at high sequencing depth (Supplementary Table S1). During the demethylation phase, we conducted RRBS on two independent preparations of PGCs with good reproducibility, and we separated male and female PGCs starting from E12.5 (Supplementary Fig S1b). Confirming previous data[10–13], we found that: PGCs were hypomethylated at E9.5 compared to the epiblast; 5mC levels were relatively stable between migratory (E9.5) and early gonadal (E10.5)

PGCs; PGCs underwent global demethylation between E10.5 and E13.5; male PGCs regained global methylation after E14.5 while female PGCs remained hypomethylated (Fig. 1a). Fetal prospermatogonia also gained non-CG methylation after E14.5 (Supplementary Fig S1c), which occurred mostly at CpA sequences and correlated with the presence of high CpG methylation (Supplementary Fig S1d, e). CpA methylation was also detected in neonatal prospermatogonia[34] but was no longer detectable in mature spermatozoa (Supplementary Fig S1c), indicating that CpA methylation persists in mitotically arrested prospermatogonia before disappearing during spermatogenesis.

We then investigated whether the kinetics of demethylation may vary among genomic sequences during PGC reprogramming. In agreement with previous studies[10,11], we observed that imprinted germline differentially methylated regions (gDMRs) showed delayed kinetics of demethylation compared to the whole genome (Supplementary Fig S1f–h). Likewise, we found that several methylated CpG island (CGI) promoters of germline genes showed a delay of demethylation compared to the genome (Fig. 1b, c). In summary, this data provides a quantitative atlas of DNA methylation reprogramming and delayed demethylation of germline genes during PGC development in the mouse.

### Site-specific resistance to DNA methylation erasure at retro-transposons in PGCs
To investigate which sequences retain DNA methylation in PGCs, we reanalyzed whole genome datasets from E13.5 PGCs[21] and uncovered 20255 and 19262 residually methylated regions (RMRs) with at least 4 CpGs and 30% methylation in female and male E13.5 PGCs, respectively (Fig. 1d). These RMRs have an average size of 1626 bp in female E13.5 PGCs and 1724 bp in male E13.5 PGCs. RMRs overlapped between male and female PGCs (Fig. 1e) and were strongly enriched in retrotransposons, mostly evolutionarily young and potentially transcriptionally active long terminal repeats (LTR) and non-LTR retrotransposons (ERVK, ERV1 and L1Md families, Fig. 1f). Detailed quantification revealed high DNA methylation at ERVK and ERV1 families, with Intracisternal A-particle (IAP) families showing the highest DNA methylation in E13.5 PGCs (40-60% at IAPs compared to 5-7% in the whole genome) (Fig. 1g, h, Supplementary Table S2), confirming the ability of evolutionarily young retrotransposons to resist DNA methylation erasure in PGCs[10–12,21,22]. Temporal RRBS analysis showed that these young retrotransposons resist the initial wave of demethylation between E9.5 and E12.5 and maintain constant DNA methylation at all stages of PGC development (Fig. 1i), which is consistent with a selective maintenance mechanism during PGC reprogramming rather than de novo activity.

To validate that locus-specific DNA methylation in PGCs requires DNMT1, we generated a conditional knock-out (cKO) of *Dnmt1* by crossing *Dnmt1−2lox* mice[35] with the *Tnap*-Cre line expressing the Cre recombinase in PGCs starting from E9.5[36] (Supplementary Fig S2a, b). PGCs from E13.5 gonads were recovered by fluorescence-activated cell sorting (FACS) with the germ cell surface marker SSEA-1 (Supplementary Fig S2c, d) and PCR genotyping confirmed the Cre-mediated deletion of *Dnmt1* in E13.5 PGCs (Supplementary Fig S2e). In agreement with a former study using *Blimp1*-Cre-mediated conditional inactivation of *Dnmt1* in PGCs[24], we observed reduced PGC numbers recovered from *Dnmt1* cKO compared to control littermate embryos (Supplementary Fig S2f). We then quantified the methylation of RMRs in mutant PGCs by RRBS, which showed that the CpGs in RMRs were hypomethylated in *Dnmt1* cKO PGCs (Fig. 1j). Hence, the methylation of young retrotransposon families was strongly reduced in *Dnmt1* cKO PGCs (10-20% in *Dnmt1* cKO PGCs compared to 35-60% in control PGCs) (Fig. 1k and Supplementary Fig S2g). The remaining methylation in *Dnmt1* cKO PGCs was probably caused by the incomplete action of the *Tnap*-Cre recombinase (Supplementary

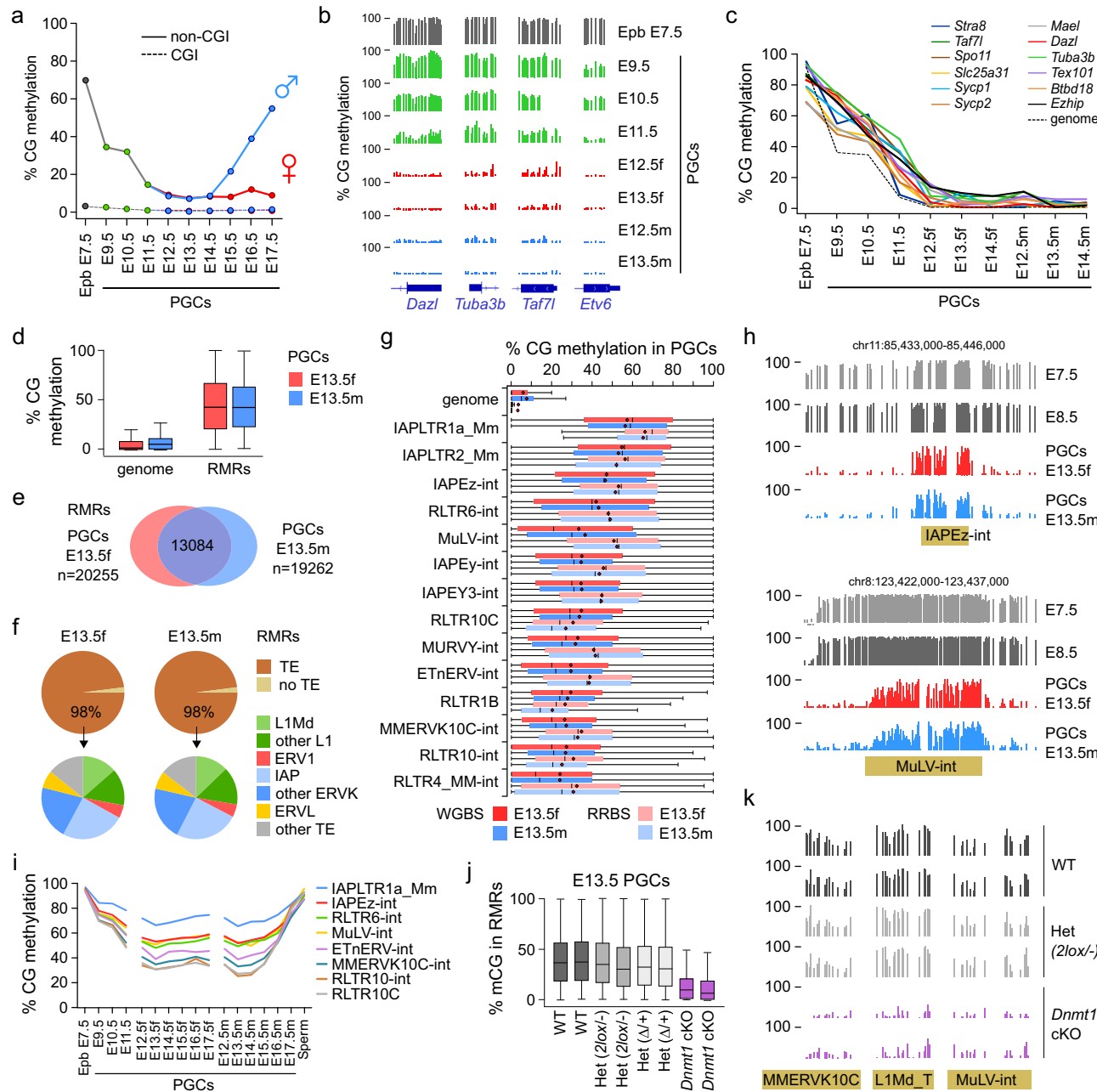

**Fig. 1 | Selective maintenance of DNA methylation in PGCs requires DNMT1.**
**a** Global CG methylation levels measured by RRBS in E7.5 epiblast (Epb) and throughout PGC development. Methylation is represented separately for CGs in CpG islands (CGI, dashed lines) and outside of CpG islands (non-CGI, full lines). The average number of covered CpGs per sample is $n = 599,050$ for CGI and $n = 560,437$ for non-CGI. **b** RRBS methylation profiles in CG-rich promoters of three germline genes (*Dazl*, *Tuba3b*, *Taf7l*), and as a control in an exonic sequence of the *Etv6* gene (chr6:134,266,250-134,266,500). Each bar represents the methylation of one CG dinucleotide. **c** Kinetics of methylation erasure in PGCs in the promoters of germline genes compared to the whole genome. For the whole genome, we represent the median methylation of all CGs with a methylation > 50% in epiblast and <20% in E13.5 PGCs. **d** Boxplots of methylation levels of individual CpGs within residually methylated regions (RMRs) compared to the whole genome in female and male E13.5 PGCs ($n = 18,909,193$ CpGs for the whole genome, $n = 350,807$ CpGs for RMRs). **e** Venn diagram of the overlap between RMRs in female and male E13.5 PGCs. **f** Pie charts showing the distribution of various transposable element (TE) families in RMRs. **g** Methylation of the top methylated retrotransposon families compared to the whole genome in E13.5 PGCs. The boxplots show the distribution

of methylation levels of individual CpGs overlapping with individual copies of each retrotransposon family in the WGBS or RRBS datasets. On average, 69% and 12% of the total number of individual genomic copies are covered in the WGBS and RRBS datasets, respectively. The numbers of CpGs and individual copies covered in each dataset are given in the source data file. **h** Examples of WGBS methylation profiles of retrotransposons carrying persistent methylation in E13.5 PGCs. **i** Quantification of CG methylation levels by RRBS for top methylated retrotransposon families throughout PGC development. **j** Boxplots of methylation levels of individual CpGs within RMRs measured by RRBS in *Dnmt1* cKO E13.5 PGCs compared to littermate controls (WT $n = 35024$ CpGs, WT $n = 33610$ CpGs, Het $n = 32389$ CpGs, Het $n = 31170$ CpGs, Het $n = 26439$ CpGs, Het $n = 23172$ CpGs, cKO $n = 27940$ CpGs, cKO $n = 24732$ CpGs). **k** Examples of RRBS methylation profiles of retrotransposons in *Dnmt1* cKO and control E13.5 PGCs (MMERVK10C chr18: 68,779,000 − 68,779,300; L1Md_T chr8: 91,423,000 − 91,423,500; MuLV-int chr8:123,427,900-123,428,800). Boxplots: center line indicates the median, red dot indicates the mean, box limits indicate upper and lower quartiles, whiskers extend to 1.5 interquartile range. Source data are provided as a Source Data file.

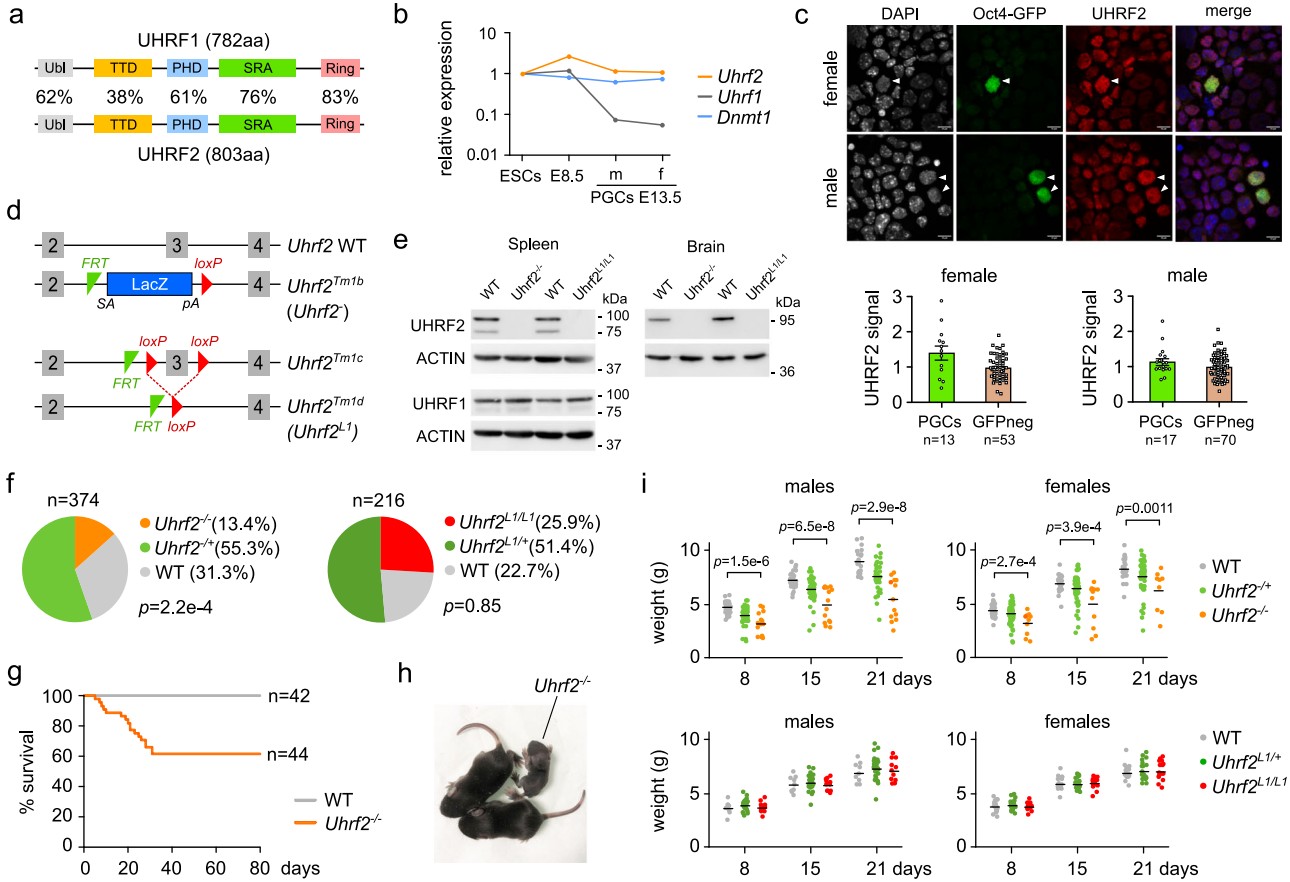

**Fig. 2 | Expression of Uhrf2 in PGCs and characterization of Uhrf2 knockout mice. a** Structure of the mouse UHRF1 and UHRF2 proteins. The percentage of sequence identity for each domain is indicated. **b** Expression of *Uhrf2*, *Uhrf1* and *Dnmt1* genes measured by RNA-seq in ESCs, E8.5 embryos, and male (m) and female (f) E13.5 PGCs (represented as a fold change relative to ESCs). Raw FPKM values are provided in the source data file. **c** Immunofluorescence analysis of UHRF2 in cells from E13.5 female or male gonads expressing *Oct4*-GFP. DNA was stained with DAPI and an anti-GFP antibody was used to identify GFP-positive PGCs. Scale bars: 10 µM. PGCs are highlighted by white arrowheads. The graphs show quantifications of UHRF2 signal normalized to GFP-negative cells (mean ± SEM, the sample sizes are indicated below the graphs). **d** Schematic representation of the *Uhrf2* knockout alleles used in the study. **e** Western blot of UHRF2 and UHRF1 proteins in spleen and brain from PND10 *Uhrf2*-deficient and WT mice. ACTIN was used as a loading control. The experiment was repeated independently three times. **f** Distribution of genotypes among 8-day-old pups obtained from *Uhrf2*^/+^ × *Uhrf2*^/+^ (left) or *Uhrf2*^L1/+^ × *Uhrf2*^L1/+^ (right) crosses (*p*-values: Chi square tests compared to the expected 1:2:1 mendelian distribution). **g** Postnatal survival curves of *Uhrf2*^-/-^ compared to littermate WT mice. **h** Photograph of an *Uhrf2*^-/-^ mouse showing growth retardation at PND10 compared to its littermates. **i** Body weights of *Uhrf2*^-/-^ mice compared to heterozygous and WT littermates obtained from crosses between *Uhrf2*^/+^ parents (top), and *Uhrf2*^L1/L1^ mice compared to heterozygous and WT littermates obtained from crosses between *Uhrf2*^L1/+^ parents (bottom). Body weights were measured 8, 15 and 21 days after birth. The bars represent the mean of the distributions. *p*-values: two-sided Mann-Whitney tests. Source data are provided as a Source Data file.

Fig S2e). Altogether, this demonstrates that DNMT1 is required for persistent DNA methylation in PGCs.

## *Uhrf2* is required for persistent DNA methylation in PGCs

Previous studies showed that UHRF1, the essential cofactor of DNMT1, is downregulated and sequestered in the cytoplasm of PGCs[11,14,15], questioning how DNA methylation is maintained at specific sequences in PGCs. We hypothesized that UHRF2, the paralog of UHRF1 (Fig. 2a), could participate in maintaining DNA methylation in PGCs. In striking contrast to *Uhrf1*, we found that *Uhrf2* expression was not downregulated in PGCs compared to ESCs and embryos (Fig. 2b). Furthermore, the UHRF2 protein was abundantly detected by immunostaining in the nucleus of E13.5 PGCs (Fig. 2c and Supplementary Fig S4a).

To study the role of UHRF2, we obtained an *Uhrf2* knock-out mouse line from the EUCOMM project (hereafter called *Uhrf2*^-/-^) in which the exon 3 is replaced by a LacZ cassette and a transcriptional termination polyadenylation sequence (*tm1b* allele) (Fig. 2d). This mutant allele leads to a frameshift knockout and reduced transcription

of the exons downstream of the cassette as demonstrated by RNA-seq in E8.5 embryos (Supplementary Fig S3a). Using antibodies specific for UHRF2 and UHRF1 in Western blotting (Supplementary Fig S4b), we confirmed that the knock-out resulted in no detectable UHRF2 protein in spleen and brain with no impact on UHRF1 in spleen (Fig. 2e). *Uhrf2*^-/-^ mice were born at sub-mendelian ratio (Fig. 2f). Furthermore, 40% of born *Uhrf2*^-/-^ mice showed growth retardation and died within 4 weeks after birth whereas the others were viable with no apparent phenotype (Fig. 2g–i). Because developmental defects were not observed in previous *Uhrf2* knockout mouse models[31,32], this prompted us to further explore this question by generating a second mouse model with a deletion of the exon 3 leading to a frameshift knockout (Tm1d allele, hereafter called *Uhrf2*^L1/L1^, see methods) (Fig. 2d, e). In contrast to *Uhrf2*^-/-^ mice, *Uhrf2*^L1/L1^ mice were born at mendelian ratio (Fig. 2f) and were viable with no growth defects after birth (Fig. 2i). One potential explanation for these differences could be linked to the fact that the *Uhrf2* gene produces transcripts that do not contain exon 3 in embryos (Supplementary Fig S3b), suggesting potentially complex roles of *Uhrf2* isoforms in development.

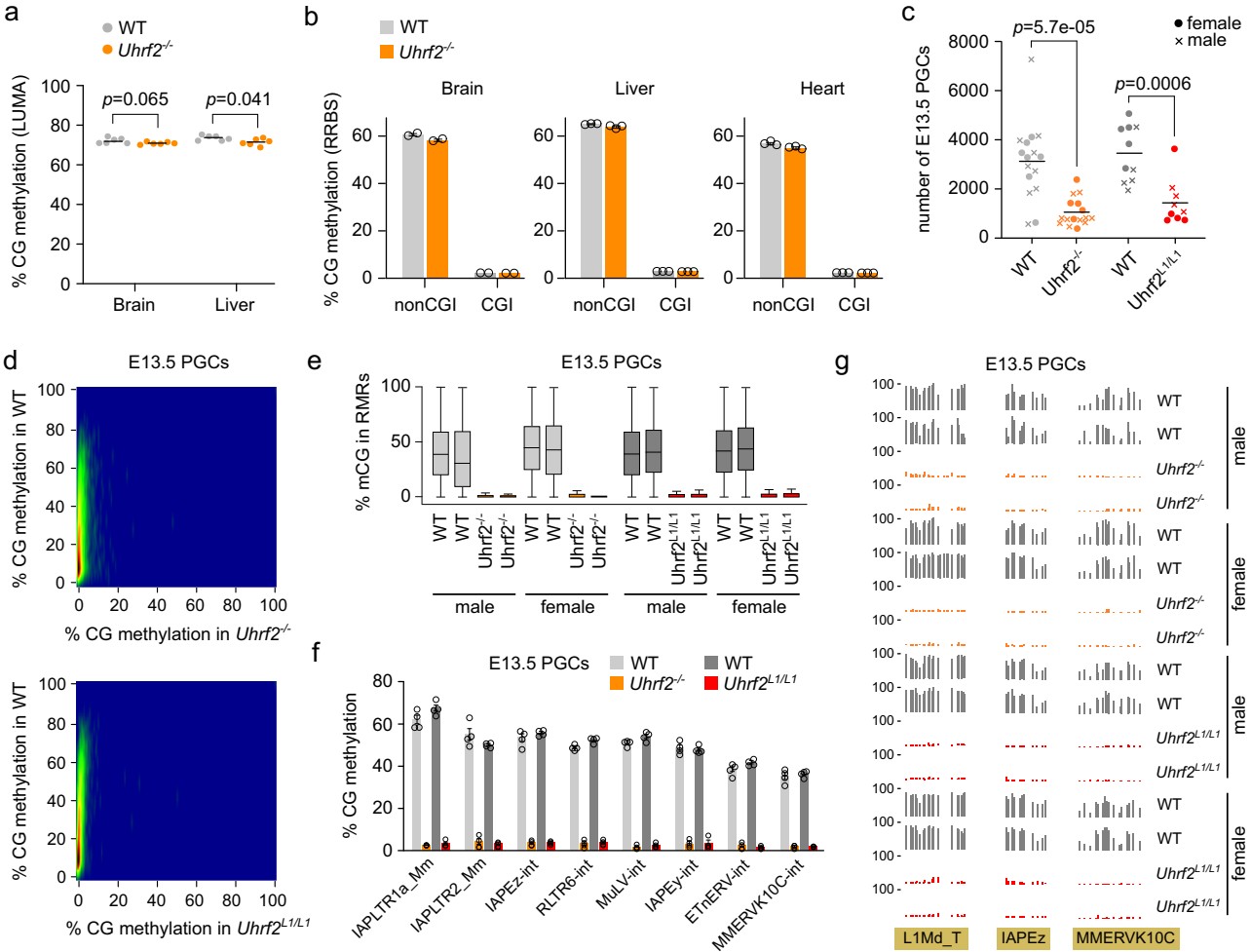

**Fig. 3 | Uhrf2 is required for DNA methylation in PGCs but not in somatic cells.**
**a** Global CG methylation levels measured by LUMA in brain and liver of *Uhrf2⁻/⁻*
compared to WT PND21 animals (*n* = 6 animals per genotype). Horizontal bars:
mean, *p*-values: two-sided Mann-Whitney tests. **b** Quantification of CG methylation
by RRBS in brain, liver and heart of *Uhrf2⁻/⁻* compared to WT PND21 animals.
Methylation levels are shown separately for CGs outside of CpG islands (non CGI) or
in CpG islands (CGI). Data are presented as mean ± SEM (*n* = 2 animals per genotype
for brain, *n* = 3 animals for liver and heart). **c** Number of PGCs recovered by flow
cytometry from gonads of *Uhrf2*-mutant compared to littermate WT E13.5 embryos
(WT *n* = 16 embryos, *Uhrf2⁻/⁻* *n* = 16, WT *n* = 10, *Uhrf2^{L1/L1}* *n* = 9). Horizontal bars:
mean, *p*-values: two-sided Mann–Whitney tests. **d** Correlation of RRBS CG methy-
lation scores in 500 bp windows in E13.5 PGCs from *Uhrf2⁻/⁻* and *Uhrf2^{L1/L1}* compared
to their littermate WT embryos (average of *n* = 4 embryos per genotype). Only tiles
with a methylation above 5% in WT PGCs are shown in these graphs. **e** Boxplots of

methylation levels of individual CpGs within RMRs in male and female *Uhrf2⁻/⁻* and
*Uhrf2^{L1/L1}* compared to WT E13.5 PGCs. Each boxplot represents an independent
animal (male WT *n* = 43422 CpGs, WT *n* = 27170 CpGs, *Uhrf2⁻/⁻* *n* = 32317 CpGs,
*Uhrf2⁻/⁻* *n* = 41483 CpGs, female WT *n* = 46505 CpGs, WT *n* = 41505 CpGs, *Uhrf2⁻/⁻*
*n* = 32272 CpGs, *Uhrf2⁻/⁻* *n* = 21081 CpGs, male WT *n* = 32084 CpGs, WT *n* = 34058
CpGs, *Uhrf2^{L1/L1}* *n* = 33392 CpGs, *Uhrf2^{L1/L1}* *n* = 32894 CpGs, female WT *n* = 33909
CpGs, WT *n* = 34215 CpGs, *Uhrf2^{L1/L1}* *n* = 31816 CpGs, *Uhrf2^{L1/L1}* *n* = 26752 CpGs).
Boxplots: center line indicates the median, box limits indicate upper and lower
quartiles, whiskers extend to 1.5 interquartile range. **f** Mean methylation levels of
ERV families in male and female *Uhrf2⁻/⁻* and *Uhrf2^{L1/L1}* compared to WT E13.5 PGCs
(mean ± SEM, n = 4 embryos per genotype). **g** Examples of RRBS methylation
profiles of retrotransposons in *Uhrf2*-mutant and WT E13.5 PGCs (L1Md_T chr8:
91,423,000−91,423,500; IAPEz chr18: 55,320,600−55,320,800; MMERVK10C chr16:
31,218,700−31,219,000). Source data are provided as a Source Data file.

We then investigated the role of UHRF2 in DNA methylation in
*Uhrf2⁻/⁻* mice. RRBS and RNA-seq analysis in *Uhrf2⁻/⁻* E8.5 embryos
revealed no difference in DNA methylation patterns and no differ-
entially expressed genes compared to WT embryos (Supplementary
Fig S3c, d, Supplementary Table S3). We then quantified DNA
methylation in organs from *Uhrf2⁻/⁻* animals at postnatal day (PND)
21, focusing on individuals that showed growth retardation (Sup-
plementary Fig S3e). We observed marginal hypomethylation in the
brain and liver of *Uhrf2⁻/⁻* animals compared to WT by performing
Luminometric Methylation Assay (LUMA) (Fig. 3a). Furthermore,
RRBS in the brain, heart, and liver revealed no hypomethylation and
very few hypomethylated differentially methylated regions (DMRs)
in *Uhrf2⁻/⁻* animals (Fig. 3b and Supplementary Fig S3f, g). We con-
clude that *Uhrf2* is largely dispensable for somatic DNA methylation
during development.

Next, we investigated the role of *Uhrf2* in DNA methylation in
PGCs. First, we examined the expression of UHRF1 by immunostaining
in WT and *Uhrf2* mutant gonads at E13.5. Similar to the WT condition,
UHRF1 was detected in the surface epithelium and in some interstitial
cells of the ovaries and testes but remained undetectable in PGCs of
*Uhrf2* knockout gonads (Supplementary Fig S5), indicating that there is
no compensatory ectopic expression of UHRF1 upon loss of UHRF2 in
PGCs. *Uhrf2* knockout lines were crossed with the *Oct4(ΔPE)*-GFP line to
isolate PGCs from E13.5 *Uhrf2⁻/⁻* and *Uhrf2^{L1/L1}* gonads. Interestingly, we
recovered fewer PGCs from E13.5 *Uhrf2⁻/⁻* or *Uhrf2^{L1/L1}* embryos com-
pared to littermate controls (Fig. 3c). RRBS analysis showed that all the
sequences with high residual DNA methylation in WT PGCs were
devoid of methylation in PGCs of *Uhrf2⁻/⁻* or *Uhrf2^{L1/L1}* male and female
embryos (Fig. 3d, e). Consistently, all retrotransposon families that
resist methylation erasure in WT PGCs were strongly hypomethylated

in male and female *Uhrf2*-deficient PGCs (2-4% in *Uhrf2*-deficient PGCs compared to 35-60% in control PGCs) (Fig. 3f, g). Global DNA methylation and TE methylation were not affected in gonadal GFP-negative somatic cells (Supplementary Fig. S3h), confirming that the requirement of UHRF2 for DNA methylation was specific to the germ cells. The loss of DNA methylation at IAPs in PGCs but not somatic cells of *Uhrf2*⁻/⁻ mice was further confirmed by Combined Bisulfite Restriction Analysis (Supplementary Fig. S3i). Altogether, these data reveal an indispensable role of UHRF2 in persistent DNA methylation in PGCs.

### *Uhrf2*-mediated DNA methylation is dispensable for retrotransposon repression in PGCs

To test if *Uhrf2*-dependent DNA methylation is required to silence transposable elements in PGCs, we performed RNA-seq in male and female E13.5 PGCs isolated from four independent *Uhrf2*⁻/⁻ embryos and WT controls (Supplementary Fig. S6a). Despite complete hypomethylation, PGCs from male or female *Uhrf2*⁻/⁻ embryos showed only subtle changes in the expression of TE subfamilies. No TE subfamily passed the significance criteria for overexpression in male PGCs (Fig. 4a). In female PGCs, very few TE subfamilies were significantly overexpressed (Fig. 4a), among which RLTR46A elements were the only ones carrying residual methylation in PGCs (Supplementary Table S4). Importantly, TE families with the highest DNA methylation in PGCs, including evolutionarily young ERVK (IAPs, MMERVK10C-int), ERV1 (RLTR6-int) or L1Md elements, were not significantly overexpressed in male or female PGCs (Fig. 4a). This is corroborated by the reanalysis of published RNA-seq data from *Dnmt1*-deficient PGCs[24] showing very little transcriptional deregulation of methylated TE families (Supplementary Fig S7a).

We conclude that the expression of transposable elements remains largely unaffected in E13.5 PGCs of *Uhrf2*⁻/⁻ embryos, suggesting that compensating mechanisms maintain their repression upon loss of DNA methylation in PGCs. Indeed, the reanalysis of published datasets[22,23] showed that RMRs are marked by the histone marks H3K9me3 and H3K27me3 in E13.5 PGCs (Supplementary Fig S7b) and that evolutionary young TE families including IAPs are overexpressed in PGCs mutant for *Setdb1* or *Ezh2* (Supplementary Fig S7c, d), which substantiates the idea that these chromatin marks compensate for the absence of DNA methylation in PGCs.

### *Uhrf2* inactivation leads to overexpression of meiotic genes in female PGCs

Next, we used the RNA-seq data to investigate if the loss of UHRF2 impacts gene expression in PGCs. The inactivation of *Uhrf2* did not lead to reduced expression of *Dnmt* or *Uhrf1* genes in PGCs (Fig. 4b). Differential expression analysis identified a 30-fold higher number of misregulated genes in female compared to male *Uhrf2*⁻/⁻ PGCs, with the majority of genes being upregulated (Fig. 4c and Supplementary Fig S6b). We also noticed significantly reduced expression of key regulators of pluripotency and PGC specification such as *Pou5f1/Oct4*, *Nanog*, *Prdm1/Blimp1*, *Tfap2c*, *Esrrb*, *Prdm14* and *Dppa3* in female *Uhrf2*⁻/⁻ PGCs, indicating compromised PGC identity in these cells (Fig. 4d). Gene ontology analysis revealed that the genes upregulated in female *Uhrf2*⁻/⁻ PGCs were strongly enriched for gene functions related to meiosis and gamete generation (Fig. 4e). Indeed, over 50 known germline and meiotic genes were overexpressed in female *Uhrf2*⁻/⁻ PGCs at E13.5 (Supplementary Table S4), including genes coding for synaptonemal complex proteins (SYCP) or STRA8, the master regulator of meiosis initiation (Fig. 4f and Supplementary Fig S6c, d). Some meiotic genes also show a minor trend of upregulation in male PGCs but at very low levels and mostly below the significance threshold (Supplementary Fig S6c), which likely reflects that male PGCs are still blocked for meiosis at this stage.

Many of these germline genes are repressed by DNA methylation in embryos (Supplementary Fig S6c) and undergo delayed demethylation in PGCs as shown in Fig. 1c, for example *Stra8*, *Sycp1*, *Sycp2*, *Taf7l*, *Slc25a31*, *Tuba3b*, *Btbd18* and *Tex101*, suggesting that UHRF2 prevents early demethylation and activation of meiotic genes in female PGCs. A similar role was proposed for *Dnmt1*[24], which predicts that the genes upregulated in *Uhrf2*⁻/⁻ PGCs overlap with the genes upregulated in *Dnmt1*-deficient PGCs. To test this hypothesis, we reanalyzed the RNA-seq data from *Dnmt1*-cKO PGCs[24] and found a highly significant overlap between the genes upregulated in *Uhrf2*⁻/⁻ and *Dnmt1* cKO female PGCs (Fig. 4g). This overlapping set of genes was strongly enriched for GO terms related to meiosis (Fig. 4g). The genes upregulated only in *Uhrf2* KO PGCs could reflect indirect effects or other functions of UHRF2 unrelated to DNA methylation. Furthermore, bisulfite sequencing analysis of the *Tuba3b* gene promoter revealed that it is prematurely demethylated in *Uhrf2*-deficient E11.5 PGCs compared to littermate controls (Fig. 4h). Altogether, these results show that the inactivation of *Uhrf2* causes premature demethylation of germline genes in PGCs leading to overexpression of a number of meiotic genes in female PGCs.

Interestingly, a significant proportion of meiotic genes upregulated by mutation of *Uhrf2* are commonly upregulated in female PGCs mutant for *Setdb1*, *Ezh2* or *Rnf2*[22,23,37] (Supplementary Fig S8), which indicates a convergence of different epigenetic pathways to regulate the timing of expression of meiotic genes in female PGCs.

The overexpression of key meiotic genes suggests that mutant female PGCs could enter meiosis precociously, which we investigated by immunostainings in E13.5 gonads. In *Uhrf2* mutant E13.5 ovaries, several STRA8-positive cells were observed, indicating they are meiotic, whereas such STRA8-positive cells were rarely detected in WT littermates (Fig. 4i). In agreement with the conclusion that germ cells had initiated meiosis at an earlier stage, many germ cells were REC8-, SYCP3- or γH2AX-positive at E13.5 in *Uhrf2* mutant ovaries, while they were scarce or absent in control ovaries (Supplementary Fig S9). In contrast, meiotic germ cells were never observed in *Uhrf2*-deficient testes at E13.5 (data not shown). These results confirm a role of UHRF2 in regulating meiotic initiation in female PGCs.

### *Uhrf2* is required for oocyte development

We subsequently analyzed the consequences of *Uhrf2* inactivation on female gamete development. Histological examination of ovary sections at PND80 revealed a reduced number of primary, secondary, and antral follicles containing oocytes in *Uhrf2*⁻/⁻ or *Uhrf2*^L1/L1 females compared to WT controls (Fig. 5a-b). Histological analysis of serial sections of PND15 ovaries also revealed a reduced number of ovarian follicles and a strongly reduced primordial follicle pool in *Uhrf2*^L1/L1 females compared to WT controls (Fig. 5c). Thus, our results show that *Uhrf2* deficiency is associated with a reduced pool of primordial follicles and reduced number of ovarian follicles in aged ovaries, indicating compromised oogenesis in the embryo.

To assess the consequences on fertility, we crossed PND65 *Uhrf2*-deficient and control females with WT males and counted the number of pups at birth. While control females gave birth to 7 pups on average, *Uhrf2*⁻/⁻ and *Uhrf2*^L1/L1 females did not give birth to pups or in rare cases to 1 or 2 pups only (Fig. 5d), indicating a severe subfertility phenotype. The severe subfertility of *Uhrf2*-deficient females suggests that most of the remaining oocytes in *Uhrf2*-deficient ovaries are not competent for embryonic development. Taken together, these results demonstrate that *Uhrf2* deficiency compromises oocyte development and female fertility in the mouse.

### *Uhrf2* inactivation impairs retrotransposon remethylation during spermatogenesis

In males, histological analysis revealed that about 10% of the seminiferous tubules were empty in testes of *Uhrf2*⁻/⁻ or *Uhrf2*^L1/L1 males (Fig. 5e), which could be a consequence of the reduced number of PGCs. Accordingly, the size of testis was slightly but significantly

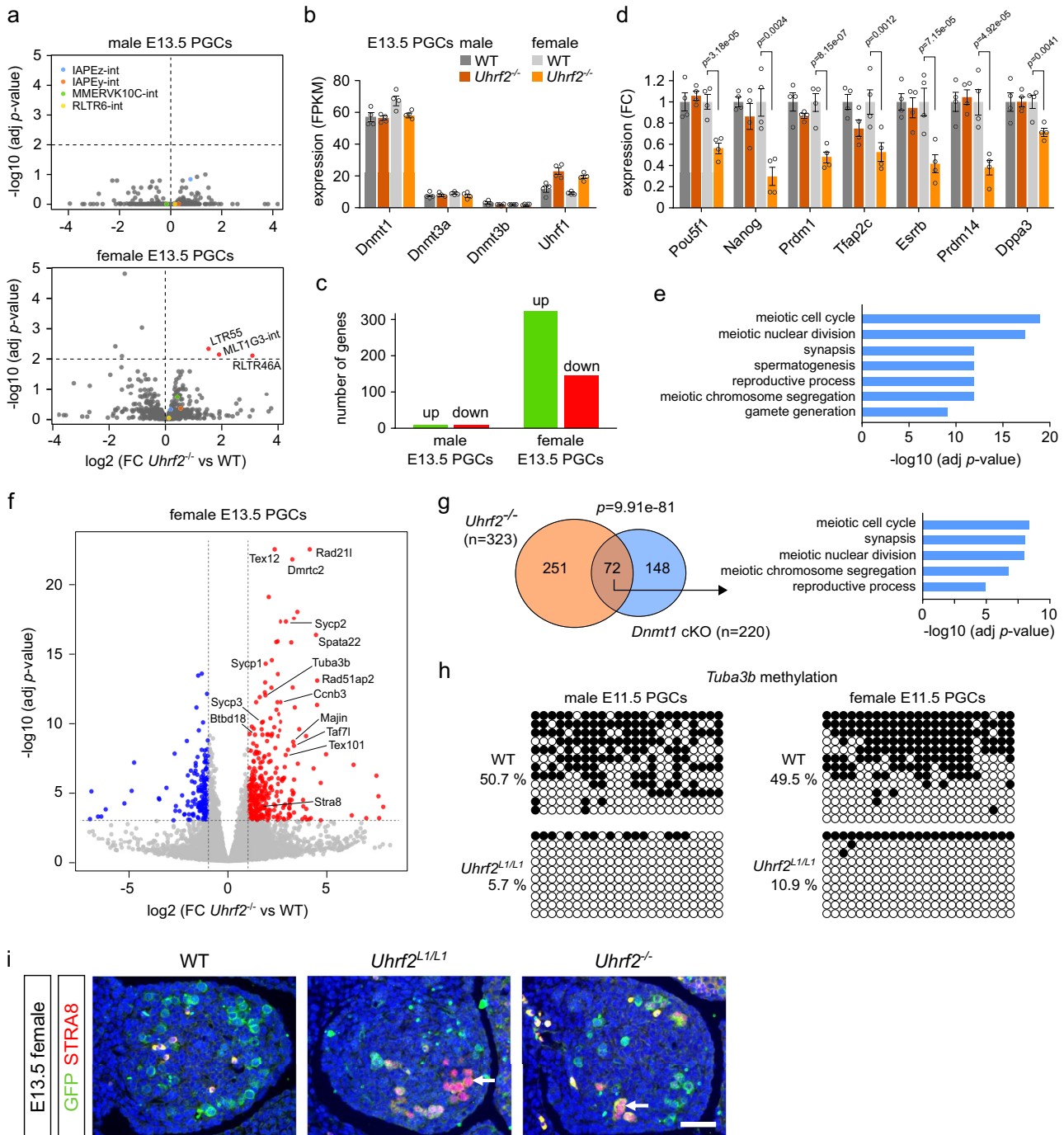

**Fig. 4 | Uhrf2 inactivation leads to increased expression of meiotic genes but not retrotransposons in E13.5 PGCs. a** Volcano plots showing differential expression of TE families in male and female *Uhrf2*[-/-] E13.5 PGCs. Significantly upregulated TE families (padj<0.01) are highlighted in red. *P*-values: DESeq2 adjusted *p*-values. **b** FPKM expression values of *Dnmt* and *Uhrf1* genes in *Uhrf2*[-/-] compared to WT E13.5 PGCs (mean ± SEM, *n* = 4 animals). **c** Number of significantly upregulated and downregulated genes identified in male and female *Uhrf2*[-/-] E13.5 PGCs. **d** Expression of regulators of pluripotency and germline identity in *Uhrf2*[-/-] compared to WT E13.5 PGCs (fold change relative to the WT of the same sex, mean ± SEM, *n* = 4 animals, *p*-values: DESeq2 adjusted *p*-values). The legend is the same as in **b**. **e** Gene ontology terms significantly enriched among genes upregulated in female *Uhrf2*[-/-] E13.5 PGCs. **f** Volcano plot showing differential gene expression in female *Uhrf2*[-/-] compared to WT E13.5 PGCs. Significantly upregulated and downregulated genes are highlighted in red

and blue respectively. The names of some upregulated meiotic genes are indicated. *P*-values: DESeq2 adjusted *p*-values. **g** Venn diagram showing the overlap between the genes upregulated in *Uhrf2*[-/-] and *Dnmt1*-cKO E13.5 PGCs. *p*-value: hypergeometric test. Gene ontology terms significantly enriched among common genes are shown on the right. **h** Bisulfite sequencing analysis of the *Tuba3b* promoter in *Uhrf2*[L1/L1] compared to WT E11.5 PGCs. Each row represents a sequenced clone (white dots: unmethylated CpGs, black dots: methylated CpGs). **i** Detection of meiotic cells by immunostaining of STRA8 (red nuclear signal) on ovarian sections from control (WT) and mutant E13.5 fetuses, as indicated. Immunostaining of Oct4-GFP (green signal) was used to detect the PGCs. Nuclei are counterstained with DAPI (blue signal). The white arrows point to STRA8-positive, meiotic, germ cells. The experiment was repeated independently at least on four gonads for each genotype. Scale bar: 30 μm. Source data are provided as a Source Data file.

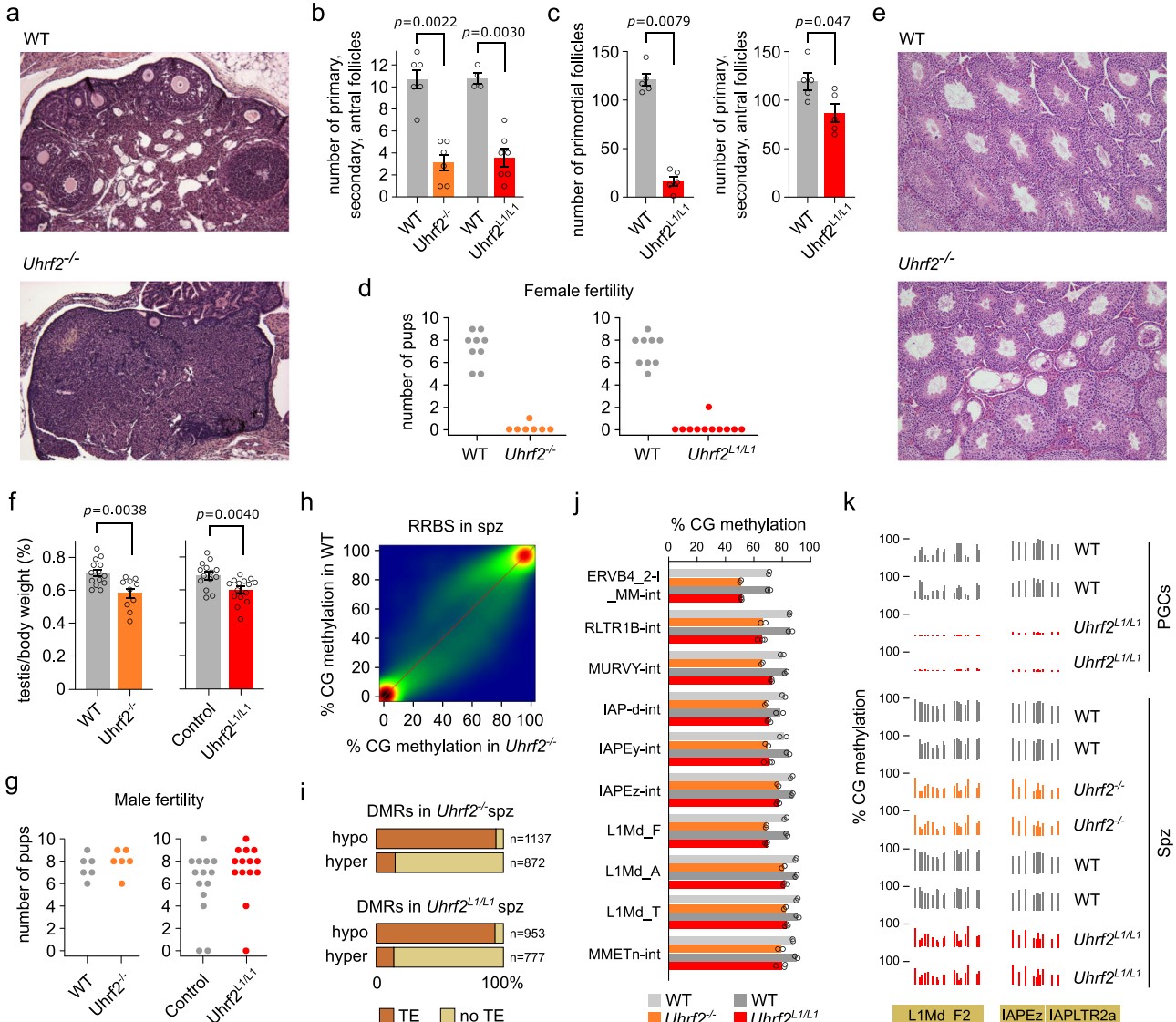

**Fig. 5 | Uhrf2 inactivation impairs oocyte development and retrotransposon methylation in sperm. a** Histological sections of ovaries stained by H&E in a WT and *Uhrf2⁻/⁻* female at PND80. The experiment was repeated independently on six gonads for each genotype. **b** Number of primary, secondary and antral follicles counted in sections of PND80 ovaries from *Uhrf2*-mutant and WT littermate females (mean ± SEM, WT *n* = 6 females, *Uhrf2⁻/⁻ n* = 6 females, WT *n* = 4 females, *Uhrf2^{L1/L1} n* = 7 females, *p*-values: two-sided Mann–Whitney tests). **c** Number of primordial follicles (left) and primary, secondary and antral follicles (right) counted in serial sections of PND15 ovaries from *Uhrf2^{L1/L1}* and WT littermate females (mean ± SEM, *n* = 5 females per genotype, *p*-values: two-sided Mann–Whitney tests). **d** Number of pups born from *Uhrf2*-deficient females compared to their WT littermate controls. **e** Representative examples of histological sections of testis stained by H&E in a WT and *Uhrf2⁻/⁻* male at PND80. The experiment was repeated independently on six gonads for each genotype. **f** Testis-to-body weight ratios in

PND80 *Uhrf2*-deficient males compared to their littermate controls (mean ± SEM, WT *n* = 14 males, *Uhrf2⁻/⁻ n* = 10 males, WT *n* = 14 males, *Uhrf2^{L1/L1} n* = 15 males, *p*-values: two-sided Welch's t-tests). *Uhrf2^{L1/L1}* males were compared to WT and *Uhrf2^{L1/+}* heterozygotes. **g** Number of pups born from crosses with *Uhrf2*-deficient males compared to their littermate controls. **h** Correlation of RRBS CG methylation scores in 500 bp windows in *Uhrf2⁻/⁻* compared to WT spermatozoa (average of *n* = 2 animals per genotype). **i** Proportion of DMRs identified in *Uhrf2*-deficient spermatozoa that colocalize with transposable elements (TEs). **j** Mean methylation levels of selected ERV and L1 families in *Uhrf2*-deficient compared to WT spermatozoa (*n* = 2 or 3 animals per genotype). **k** Two examples of sequences in retrotransposons hypomethylated in *Uhrf2*-deficient PGCs and spermatozoa (L1Md_F2 chr4: 63,921,700–63,922,300; IAP chr19: 13,269,700 – 13,269,900). Source data are provided as a Source Data file.

reduced in *Uhrf2⁻/⁻* and *Uhrf2^{L1/L1}* compared to littermate control animals (Fig. 5f). Nevertheless, the production of functional spermatozoa was not impaired and *Uhrf2* deficient males appeared fertile (Fig. 5g).

To investigate if the inactivation of *Uhrf2* has a consequence on the methylome of spermatozoa, we performed RRBS in spermatozoa isolated from *Uhrf2⁻/⁻* or *Uhrf2^{L1/L1}* males. We observed strong perturbations with both gain and loss of DNA methylation compared to spermatozoa isolated from WT littermates (Fig. 5h). These perturbations were highly reproducible between independent animals and between *Uhrf2⁻/⁻* and *Uhrf2^{L1/L1}* lines (Supplementary Fig S10a). Our

analysis of DMRs showed that most hypomethylated DMRs (hypo-DMRs) overlapped annotated transposable elements, in contrast to hypermethylated DMRs (hyper-DMRs) that mostly occurred in single-copy sequences (Fig. 5i). Sperm hypo-DMRs were frequent in evolutionarily young retrotransposon families such as L1Md, IAP and other ERVK elements (Supplementary Fig S10b), and quantification of DNA methylation of retrotransposon families confirmed a reduction of 10-15% average methylation of several ERVK (i.e., IAPs, MMERVK10C-int, MMETn-int), ERV1 (i.e., RLTR1B-int, MURVY-int) and L1Md (i.e., L1Md_F, L1Md_F2, L1Md_A, L1Md_T) families in spermatozoa from *Uhrf2⁻/⁻* and

*Uhrf2^{L1/L1}* animals (Fig. 5j, k and Supplementary Fig S10c). Furthermore, we found that sperm hypo-DMRs overlap with hypo-DMRs in *Uhrf2*-deficient PGCs, illustrating that many of the regions hypomethylated in *Uhrf2*-deficient PGCs did not recover full methylation in spermatozoa (Supplementary Fig S10d, e). In contrast, sperm hyper-DMRs were not methylated in PGCs and arose subsequently during spermatogenesis (Supplementary Fig S10e), and they were not restricted to specific sequence features (Supplementary Fig S10f). We conclude that UHRF2 is not only required to protect evolutionarily young retrotransposons from demethylation in PGCs but also to restore their full methylation during spermatogenesis. These results demonstrate that the inactivation of *Uhrf2* perturbs sperm DNA methylation but is nevertheless compatible with the production of functional spermatozoa.

## Discussion

DNA methylation erasure in primordial germ cells has been attributed to impaired maintenance activity because of downregulation and primary localization in the cytoplasm of UHRF1 in PGCs[11,14]. This left open the question of how DNA methylation is maintained at specific sequences in PGCs in the absence of UHRF1 activity. UHRF2 is a conserved paralog of UHRF1, but its biological functions have remained enigmatic. In this manuscript, we present evidence that UHRF2 has a role in regulating site-specific maintenance of DNA methylation during germ cell development. We reveal that persistent DNA methylation of retrotransposons is abolished in male or female PGCs of *Uhrf2*-KO mice. This suggests that UHRF2 could compensate for UHRF1 to recruit DNMT1 at specific sequences in PGCs. Alternatively, it cannot be ruled out that UHRF2 has an accessory function to stimulate low levels of UHRF1 still present in the nucleus of PGCs. A conditional inactivation of *Uhrf1* in early PGCs would help clarify this question. It can also not be excluded that de novo methylation activity by DNMT3 enzymes contributes to sustaining DNA methylation of retrotransposons in PGCs.

UHRF1 and UHRF2 are related proteins sharing similar domains, which questions why UHRF2 only acts at transposable elements in PGCs, whereas UHRF1 is a genome-wide maintenance factor. Transposable elements are preferentially targeted for the deposition of H3K9me3 in PGCs[23]. The TTD and PHD domains are the least conserved domains between UHRF2 and UHRF1 (Fig. 2a) and mediate higher specificity for H3K9me3 in UHRF2 compared to UHRF1[27,28,38]. Furthermore, UHRF2 shows reduced preference for hemimethylated DNA[27,28]. These divergences may explain why the activity of UHRF2 is not general but directed at H3K9me3-marked regions. Additionally, UHRF2 has been identified as a 5hmC reader through its SRA domain[29,30], and its H3 ubiquitin ligase activity is robustly stimulated by this mark[28]. 5hmC is re-localized to repetitive elements during PGC development[13] and could also contribute to specifically directing UHRF2 activity at these sites. Lastly, the difference in selectivity could be due to unknown protein partners that target the action of UHRF2 in PGCs.

Our results reveal that the inactivation of *Uhrf2* impairs retrotransposon methylation not only in PGCs but also in mature sperm. Two explanations can be proposed to account for this observation: either methylation cannot be fully restored in sperm when starting from null methylation in PGCs because of the limited amounts of proteins required for de novo methylation, such as DNMT3C[2], or UHRF2 continuously participates in maintenance DNA methylation at ERVs during spermatogenesis alongside UHRF1[39,40]. Since methylation of retrotransposons is also reduced in late male germ cells mutant for piRNA biogenesis factors[41,42], it is possible that UHRF2 is required for maintenance of DNA methylation following piRNA-dependent de novo DNA methylation. Moreover, the discovery of hyperDMRS in spermatozoa is very puzzling considering the potential maintenance function of UHRF2. It is possible that the chromatin landscape and recruitment of de novo methylation machinery are modified at some sequences in the absence of UHRF2, or that the UHRF1/DNMT1 axis is stimulated in

the absence of UHRF2 leading to excessive maintenance methylation at some sequences during spermatogenesis.

The biological reasons why DNA methylation persists at evolutionarily young retrotransposons in PGCs remain unclear. We detected no major upregulation of LINE or ERV expression in E13.5 PGCs of *Uhrf2*-KO animals, which suggests that persistent DNA methylation is not required to maintain the repression of retrotransposons in PGCs. Yet, the possibility remains that retrotransposons could be upregulated at other time points of PGC development. Our results are consistent with a previous study showing that DNMT1 is not required for the repression of IAP and L1 retrotransposons in E13.5 PGCs[24], and point to compensating mechanisms maintaining the repression of potentially active and harmful retrotransposons during epigenetic reprogramming in PGCs. In PGCs, de novo H3K9me3 peaks occur specifically at ERVs in E13.5 PGCs, and young ERVs are reactivated upon conditional knockout of *Setdb1* in PGCs[22,23] (Supplementary Fig S7c), suggesting that H3K9me3 provides repression of transposable elements in the absence of DNA methylation. Additionally, recent work also showed that DNA methylation reduction is compensated by the accumulation of H3K27me3 at retrotransposons to ensure TE repression in embryonic cells and female PGCs[23,43] (Supplementary Fig S7d). In complement, it is also possible that piRNAs already present in E13.5 PGCs contribute to degrade transcripts derived from retrotransposons[42]. Thus, DNA methylation does not appear as the primary mode of TE repression in PGCs but could still serve as a backup system in case other repressive mechanisms are compromised. In males, the analysis of *Dnmt3L*-KO animals revealed that DNA methylation is dispensable for silencing retrotransposons before meiosis but prevents them from generating catastrophic meiotic recombination events[44]. By analogy, retrotransposon DNA methylation in PGCs could be required to prevent meiotic catastrophes during female meiosis, which is initiated soon after E13.5. This model deserves further investigations and is compatible with compromised oocyte development observed in *Uhrf2*-deficient females.

We also discovered that *Uhrf2* controls the level of expression of meiotic genes in PGCs. Female *Uhrf2*-KO PGCs overexpress a number of meiotic and germ-cell-specific genes, including the master regulator of meiosis initiation *Stra8*, which induces premature meiotic entry and could further contribute to compromised oocyte development. This suggests that *Uhrf2* cooperates with DNMT1 to prevent early demethylation and activation of meiotic genes in PGCs[24]. UHRF2 could also have roles in repression of germline genes independently of DNA methylation, for example, by participating in the recruitment of PRC1[37]. Interestingly, an overlapping set of meiotic genes overexpressed in *Uhrf2* mutant PGCs is also upregulated upon inactivation of *Rnf2*, *Setdb1* or *Ezh2* in PGCs (Supplementary Fig S8), which demonstrates the existence of parallel coordinated epigenetic mechanisms involving DNA methylation (UHRF2/DNMT1)[24], PRC1/RNF2[37] and PRC2 (EED/EZH2)[23,45] that converge to limit precocious expression of meiosis-related genes and control the timing of meiosis initiation in female PGCs. Similar to the inactivation of *Dnmt1*, *Rnf2*, or *Ezh2*[23,24,37], the inactivation of *Uhrf2* has a much lower impact on the expression of meiotic genes in male PGCs. This highlights that male PGCs are less sensitive to the perturbation of epigenetic pathways, presumably because they are programmed to initiate meiosis much later and therefore are protected from meiotic signals or lack transcription factors necessary to activate the meiotic program.

The biological functions of *Uhrf2* outside of germ cells remain to be clarified. We observed partially penetrant pre- and post-natal lethality associated with growth retardation in *Uhrf2^{-/-}* animals, suggesting potential roles in development. However, this deserves further clarification because *Uhrf2^{L1/L1}* mice or other previously reported *Uhrf2* knockout mouse models showed no developmental phenotype[31,32]. These differences could be attributed to indirect effects caused by the insertion cassette in *Uhrf2^{-/-}* animals or to the incomplete inactivation

of all *Uhrf2* isoforms in other KO models. It is conceivable that naturally occurring mRNA isoforms lacking exon 3, which are disrupted in the *Uhrf2[-/-]* but not in the *Uhrf2[L1/L1]* model, could produce truncated proteins from a downstream ATG. Such truncated proteins cannot be detected by the antibody used in this study because it recognizes an epitope encoded by exon 3 of the gene. Furthermore, we found no evidence of a major role of *Uhrf2* in global DNA methylation in embryos and postnatal organs, consistent with studies showing that *Uhrf2* is unable to rescue DNA methylation in *Uhrf1*-null mouse ES cells[26,27]. Perhaps UHRF2 could participate in the maintenance of DNA methylation at retrotransposons during the global reduction of genome methylation in preimplantation embryos, since UHRF1 is mainly localized in the cytoplasm at these stages[46]. This does not exclude that *Uhrf2* might be involved in the local modulation of 5mC/5hmC levels in specific cell types[31,32,47,48]. Finally, UHRF2 has been shown to interact with various chromatin regulators and transcription factors[49], and thus could also influence developmental gene expression programs independently of DNA methylation.

## Methods

### Mouse lines

Animal experimental procedures complied with ethical regulations and were approved by the local ethical committee (CREMEAS) and the government authority. PGCs were isolated using the Oct4-GFP mouse line (GOF18-Oct4ΔPE-GFP)[33]. Germline-specific *Dnmt1* knockout embryos were generated using *Dnmt1−2lox* mice (Dnmt1<tm2Jae>)[35] on a C57BL6/J background and the *Tnap*-Cre line (129-Alpltm1(cre)Nagy/J)[36] on a 129 background following the breeding scheme described in the Supplementary Fig S2b. The *Uhrf2*-Tm1b (C57BL/6N-Uhrf2<tm1b(EUCOMM)Wtsi>) and Tm1c (C57BL/6N-Uhrf2<tm1c(EUCOMM)Wtsi>) mouse lines were made as part of the NorCOMM2 project from EUCOMM ES cells[50] on a C57BL/6 N background at the Toronto Centre for Phenogenomics and obtained from the Canadian Mouse Mutant Repository. *Uhrf2*-Tm1b mice were backcrossed to C57BL/6J for a minimum of 4 generations before initiating the experiments. The *Uhrf2−*1lox mouse line was generated in the laboratory by crossing *Uhrf2*-Tm1c mice with *Tnap*-Cre mice on a C57BL/6 J background and backcrossed to C57BL/6 J for a minimum of 4 generations before initiating the experiments. *Uhrf2* homozygous KO mice and *Uhrf2[+/+]* wild type littermate mice born from *Uhrf2[-/+]* parents were used for experiments. Mice were housed with free access to food and water, a 12 h light/dark cycle, controlled temperature (20–24 °C), and humidity (30–60%).

### Isolation of embryos, PGCs, and spermatozoa

Embryos and germ cells were obtained by natural mating. The morning of the vaginal plug was designated E0.5, and all dissections were performed at 10 am. E8.5 embryos were manually dissected in PBS. For generating the methylation atlas of PGC development, C57BL/6 J females were crossed with *Oct4-GFP[1/1]* males, and PGCs from several embryos were pooled at each stage. Starting from E12.5, the sex of the embryos was determined anatomically. The inferior part of the embryo (E9.5-E11.5) or gonads (E12.5-17.5) were dissected in PBS and dissociated for 5 min at 37 °C in M2 medium (Sigma-Aldrich M7167), 0.25% Trypsin, 0.16 mg/mL DNAse, and 1/12Vo Accumax (Millipore SCR006) to obtain a single-cell suspension. The cells were filtered on a 70 μM Cell Strainer (MACS SmartStrainer, Miltenyi Biotec 130-098-462) and sorted using a FACSVantage flow cytometer (BD Biosciences) to isolate GFP-positive cells. For isolation of *Uhrf2*-deficient PGCs, *Uhrf2[-/+]* and *Uhrf2[L1/+]* females were crossed respectively with *Uhrf2[-/+]* Oct4-GFP[1/0] and *Uhrf2[L1/+]* Oct4-GFP[1/0] males, and GFP-positive cells were sorted using a FACSAria Fusion cell sorter (BD Biosciences). PGCs of *Dnmt1*-cKO embryos were isolated using the SSEA-1 (CD15) surface marker by incubating cell suspensions 10 min at 4 °C with anti-SSEA-1-PE antibodies (Miltenyi Biotec

130-104-936, 1:50) diluted in M2 medium with 4 μg/mL of DNAse I and 2% FBS, followed by cell sorting using a FACSVantage flow cytometer (BD Biosciences). For isolation of spermatozoa, the epididymis was dissected in M2 medium, incubated 5 min at 37 °C, and the epididymal fluid was squeezed out. Spermatozoa were centrifuged 5 min at 800 *g* and rinsed with PBS.

### Validation of PGC purity by phosphatase alkaline staining

An aliquot (50 μL) of cells obtained before and after sorting was centrifuged on a slide for 1 min at 500 rpm using a Shandon Cytospin 3 centrifuge. The cells were fixed for 1 min with paraformaldehyde 4% and rinsed with TBST buffer (20 mM Tris HCl, 150 mM NaCl, 0.05% Tween20). Alkaline Phosphatase was stained using the Alkaline Phosphatase Detection Kit (Millipore SCR004) following the manufacturer's instructions.

### DNA and RNA preparation

Genomic DNA samples were prepared by proteinase K digestion, phenol/chloroform/Isoamyl alcohol (25:24:1) extraction, and precipitation with ethanol. For gDNA extraction from sperm, the cells were lysed with a specific buffer (20 mM Tris PH 7, 10 mM DDT, 10 mM EDTA, 150 mM NaCl, 10 mM KCl, 1% SDS, 1 μg/mL proteinase K) before phenol/chloroform/isoamyl alcohol extraction and precipitation with ethanol. Total RNA samples were extracted using the RNeasy Micro kit (Qiagen 74004).

### Luminometric Methylation Assay (LUMA)

LUMA was performed as described[51]. Briefly, 500 ng of genomic DNA was digested with MspI+EcoRI or HpaII+EcoRI (New England BioLabs). HpaII is a methylation-sensitive restriction enzyme, and MspI is its methylation-insensitive isoschizomer. EcoRI was included for internal normalization. The extent of the enzymatic digestions was quantified by pyrosequencing (PyroMark Q24), and global CpG methylation levels were then calculated from the HpaII/MspI normalized peak height ratio.

### RRBS

RRBS libraries were prepared as described[52]. Between 2 and 100 ng of genomic DNA were digested with MspI (Thermo Fisher Scientific) for 5 h at 37 °C, end-repaired and A-tailed with 5 U Klenow fragment exo- (Thermo Fisher Scientific) for 40 min at 37 °C and ligated to methylated adapters overnight at 16 °C with 30 U T4 DNA ligase (Thermo Fisher Scientific) in Tango 1X buffer. Fragments between 150 and 400 bp were excised from a 3% agarose 0.5X TBE gel, purified with the MinElute gel extraction kit (Qiagen), and bisulfite converted with the EpiTect bisulfite kit (Qiagen) with two consecutive rounds of conversion. Final libraries were amplified with PfU Turbo Cx hotstart DNA polymerase (Agilent) (2 min at 95 °C; 14 to 18 cycles of 30 sec at 95 °C, 30 sec at 65 °C, 45 sec at 72 °C; final extension 7 min at 72 °C). Libraries were purified with AMPure XP beads (Beckman Coulter) and sequenced (2 × 75 bp or 2 × 100 bp) at Integragen SA (Evry, France). Reads were trimmed to remove low-quality bases with Trim Galore v0.4.2 and aligned to the mm10 genome with BSMAP v2.74 (parameters -v 2 -w 100 -r 1 -x 400 -m 30 -D C-CGG -n 1). Methylation scores were calculated using methratio.py in BSMAP v2.74 (parameters -z -u -g) and CpGs were filtered at a minimum of 8 X sequencing depth.

### Bioinformatic analysis of methylation data

Methylation in specific genomic features was calculated by selecting CpG positions that overlap with genomic features using the findOverlaps function of the IRanges package in R. The following genome annotations were used: genomic coordinates for imprinted germline DMRs from the Wamidex database, UCSC CpG island annotations, and UCSC Repeatmasker annotations for calculating methylation of TE families. RMRs were defined from WGBS data in male and female E13.5

PGCs[21] using 500 bp sliding windows with 250 bp offset to identify regions containing at least 4 CpGs and more than 30% residual methylation in E13.5 PGCs. We then selected all CpGs common to RMRs identified in male and female E13.5 PGCs (n = 350,807 CpGs) for further analysis. From this set, an average of $32,473 \pm 1,271$ CpGs ($9.26 \pm 0.36\%$) were covered in the various RRBS datasets generated in mutant PGCs. Correlation plots of RRBS values were generated by plotting mean CpG methylation scores in 500 bp windows. In case of replicates, we first calculated the mean methylation for all individual CpGs common to all replicates. eDMR from the methylKit R package was used to identify DMRs in somatic organs and sperm with at least 4 CpGs, a difference in methylation >20% and a $q$-value < 0.01. The sequence logo of CH methylation in E17.5 prospermatogonia was generated by extracting the sequence context of all CH sites with >2% methylation and using the seqLogo package in R.

## COBRA

Genomic DNA was converted with the EpiTect bisulfite kit (Qiagen 59104) and the GAG region of IAPEz elements was amplified by touchdown PCR with the Platinum Taq DNA Polymerase (Thermo Fisher Scientific) using the following program: 20 cycles of 30 s at 95 °C, 30 s at 60–50 °C (with a 0.5 °C decrease per cycle), 50 s at 72 °C followed by 35 cycles of 30 s at 95 °C, 30 s at 50 °C and 50 s at 72 °C. The PCR products were purified with the NucleoSpin Gel and PCR Clean-up kit (Macherey-Nagel 740609). 40 ng of PCR product were digested with 5U Taq1α (Thermo Fisher Scientific) for 30 min at 65 °C and loaded with 40 ng of undigested control on an agarose gel. The following primers were used to amplify the IAPEz GAG region: forward ATTTTGTTGATTAAATAAATTATTATTGGG, reverse TAAAACATATCCTCTAATCATTTCTACTCA.

## Bisulfite sequencing in PGCs

PGCs were isolated from E11.5 embryos as described above. For each sex and genotype, 2 independent experiments were performed on PGCs recovered from 2 or 3 embryos, and the sequencing results were merged. Genomic DNA from PGCs was extracted and bisulfite converted individually for each embryo with the EpiTect Fast LyseAll Bisulfite Kit (Qiagen 59864) following the manufacturer's instructions, and the samples were pooled on a MinElute DNA Spin Column before the desulfonation and purification steps. The *Tuba3b* promoter was amplified by touchdown PCR with the Platinum Taq DNA Polymerase (Thermo Fisher Scientific). The 50 µL PCR reaction was split into 5 tubes to reduce PCR biases and the following program was used: 20 cycles of 30 s at 95 °C, 30 s at 62–52 °C (with a 0.5 °C decrease per cycle), 50 s at 72 °C followed by 35 cycles of 30 s at 95 °C, 30 s at 52 °C and 50 s at 72 °C. The PCR products were purified with the PCR cleanup kit (Macherey Nagel), cloned by TA cloning in the pCR2.1 vector (Thermo Fisher Scientific, K204001), and sequenced by Sanger sequencing. Sequences with identical methylation patterns were removed to reduce clonal biases. The following primers were used to amplify the *Tuba3b* promoter: forward TAGATTAGGGAAGTTTGAGTATTTTATTTGTT, reverse CTCACAAAACCCCAAACTCTAAAAA.

## RNA-seq

RNA-seq libraries were prepared from total RNA of individual E8.5 embryos (3 WT and 3 *Uhrf2*-/- embryos) or PGCs from individual E13.5 embryos (4 WT and 4 *Uhrf2*-/- male embryos, 4 WT and 4 *Uhrf2*-/- female embryos). RNA integrity was checked using a Bioanalyzer (Agilent Technologies). For E8.5 embryos, RNA-seq libraries were generated from > 200 ng total RNA using the TruSeq Stranded Total RNA Library Prep kit with Ribo-Zero depletion (Illumina) according to the manufacturer's instructions. For PGCs, RNA-seq libraries were generated from 1 to 10 ng total RNA using the Ovation RNA Seq System V2 (NuGEN) according to the manufacturer's instructions. The libraries were generated and sequenced in paired-end 2 × 100 bp on an Illumina

sequencer by the GenomEast sequencing platform. Quality control was checked by FastQC and reads were mapped to the mouse mm10 genome using TopHat2 v2.0.13 with a RefSeq transcriptome index. Bigwig files were generated using bam2wig.py from the RSeQC package v2.4 (parameters -u -t 5000000000) and visualized in the Integrative Genomics Viewer (IGV). For gene expression analysis, unique reads were counted in RefSeq genes with HTSeq v0.9.1 (parameters −t exon −s reverse), and differentially expressed genes were identified with DESeq2 v1.20.0 using the following parameters: fold change > 2, adjusted $p$-value < 0.001. The FPKM values and PCA analysis were generated using DESeq2. We used the same pipeline for the reanalysis of published RNA-seq in *Dnmt1*, *Setdb1*, or *Ezh2* cKO PGCs and defined differentially expressed genes with the parameters fold change > 2 and adjusted $p$-value < 0.01. The list of differentially expressed genes identified by microarrays in *Rnf2* cKO PGCs was taken from the supplementary tables of the original article[37]. The expression of TEs was analyzed by counting unique and multiple-mapping reads in RepeatMasker TE families using featureCounts from the Rsubread package v1.30.9 with the option to weight multimapping reads by the number of mapping sites (parameters countMultiMappingReads = TRUE, fraction = TRUE, useMetaFeatures = TRUE). Differential expression of TE families was analyzed using DESeq2 v1.20.0. Volcano plots were generated with VolcanoseR (https://huygens.science.uva.nl/VolcaNoseR). Gene ontology analysis of differentially expressed genes was performed using DAVID 6.8 (https://david.ncifcrf.gov).

## Analysis of ChIP-seq data

ChIPseq reads were retrieved from GEO, mapped against the mm10 genome using bowtie v2-2.3.0, and converted to bigwig using bedtools bamTobed v2.27.1 and bedGraphToBigWig from UCSC tools. RMR regions were extended to 5000 bp from the RMR center on each side, and bigwig signals overlapping those extended regions were extracted using bwtool extract v1.0. As a control, an equal number of shuffled regions of the same size were generated with bedtools shuffle v2.27.1.

## Histology

Ovaries were fixed for 24 h at room temperature in 4% (w/v) formaldehyde's fixative solution. Testes were fixed for 16 h at room temperature in Bouin's fixative solution. The fixed tissues were rinsed three times with EtOH 70% and conserved at 4 °C in ETOH 70%. The fixed tissues were embedded in paraffin, and 5 µm-thick sections were stained with hematoxylin and eosin according to standard procedures.

## Immunofluorescence on isolated gonadal cells

Cells of E13.5 gonads dissected from *Oct4-GFP*^I/O^ animals were dissociated as described above. 50 µL of dissociated cells were loaded on a double cytofunnel (Thermo Fisher 5991039) and spun down using a Shandon Cytospin 3 Centrifuge at 500 rpm for 1 min on coated Shandon Double Cytoslides (Thermo Fisher 5991055). The slides were fixed 10 min with 4% paraformaldehyde, permeabilized 10 min with 0.2% Triton X-100 in PBS, and quenched 10 min with 50 mM NH4Cl. The cells were washed two times in PBS between each step. The cells were then incubated 1 hour in blocking solution (PBS/10% FBS) and stained for 1 hour at room temperature with the primary antibodies against GFP and UHRF2 (Supplementary Table S5). After two washing steps in PBS, the cells were incubated 30 min at room temperature with the secondary antibodies Alexa Fluor 488-conjugated donkey anti-rat IgG (Jackson ImmunoResearch 712-545-153, 1:250) or Alexa Fluor 594-conjugated donkey anti-mouse IgG (Abcam ab150108, 1:1500). Cells were mounted on microscope slides with DAPI Fluoromount mounting medium (Invitrogen 00495952). Images were captured using an inverted IX83 microscope with a 60x NA 1.2 water objective (Olympus) equipped with a CSU-W1 spinning disk head (Yokogawa) and an ORCA fusion cMOS camera (Hamamatsu). Z-stack images were acquired every 0.5 µm. Quantifications were done with ImageJ v2.14.0. All the

images were converted to 8 bit and a sum projection was done on z-stacks. The regions of interest (ROIs) were determined on the DAPI channel, and the mean intensity (MI) of UHRF2 fluorescence in each ROI was normalized by the area (A). Then, the MI/A values of each cell were normalized by the average MI/A of five GFP-negative cells in the same field of view. For the control in Supplementary Fig S4a, we performed UHRF2 immunofluorescence as described above on gonadal cells from a WT and *Uhrf2⁻/⁻* female at E13.5 and acquired images using an upright Leica DMRA2 microscope (Leica Microsystems GmbH) with a 100x NA 1.4 oil objective equipped with an ORCA-ER camera (Hamamatsu).

## Immunostaining on sections

Fetuses were fixed in 4% (w/v) paraformaldehyde (PFA) for 16 h at 4 °C, washed in PBS, dehydrated, embedded in paraffin, and 5 μm-thick sections were made. Antigens were retrieved for 1 hour at 95 °C in 10 mM sodium citrate buffer at pH 6.0. The sections were then incubated with the primary antibodies (Supplementary Table S5) diluted in PBS containing 0.1% (v/v) Tween 20 (PBST) for 16 h at 4 °C in a humidified chamber. After rinsing three times for 3 minutes in PBST, the bound antibodies were detected using Cy3-conjugated or Alexa Fluor 488-conjugated secondary antibodies, for 1 hour at room temperature (RT) in a humidified chamber. Nuclei were counterstained with 4′,6-diamidino-2-phenyl-indole (DAPI) (Roche Diagnostics, Meylan, France) diluted at 10 μg/ml in the mounting medium (Vectashield, Vector Laboratories).

## Mouse fertility tests

Fertility tests were carried out when *Uhrf2* mutant and littermate control mice were approximately 65 days old. The mice were housed with an adult C57BL/6J mouse for 14 days. Following this period, the females were isolated in a cage and monitored daily for up to 3 weeks to count the number of pups at birth.

## Culture and transfection of HEK293T cells

Plasmids coding for Myc-DDK-tagged UHRF1 and UHRF2 were purchased from Origene (MR210622 and MR210744). HEK293T cells were cultured in DMEM (4.5 g/L glucose) medium complemented with FBS and Penicillin/Streptavidin. 10 μg of each plasmid was transfected with jetPEI (PolyPlus 101-10 N) following the manufacturer's protocol. Protein extracts were recovered 24 h after transfection for Western Blot analysis.

## Western blots

Total protein extracts were run on an SDS PAGE gel and transferred to a 0.2 μm nitrocellulose membrane. The membrane was blocked with PBS, 0.1% Tween-20, 5% milk for 2 h at room temperature and incubated with the primary antibodies overnight at 4 °C. The membrane was washed three times, incubated with peroxidase-conjugated secondary antibody for 1 h at room temperature, and washed three times. The signal was detected by chemiluminescence using the ECL detection reagent (Amersham, GE Healthcare). The following primary antibodies were used: UHRF1 (H8, Santa Cruz sc-373750, 1:100), UHRF2 (C-10, Santa Cruz sc-398953, 1:100), FLAG (Origen TA50011-100, 1:2000), α-TUBULIN (Sigma Aldrich T9026, 1:1000) and ACTIN (Sigma Aldrich A2066, 1:10000).

## Statistical analysis

Chi-square and hypergeometric tests were performed in R. Statistical analyses to compare two sample distributions were performed using GraphPad Prism v8.0.2. If the data passed all four normality tests (D'Agostino-Pearson test, Shapiro-Wilk test, Anderson-Darling test, and Kolmogorov-Smirnov test), an unpaired two-tailed Student's t-test with Welch's correction was used. For data not passing normality tests, exact *p*-values calculated by Mann–Whitney tests are reported. In all figures, data are represented as individual data points or mean ± standard error of the mean (SEM).

## Reporting summary

Further information on research design is available in the Nature Portfolio Reporting Summary linked to this article.

## Data availability

The sequencing datasets generated in this study (RRBS and RNA-seq) have been deposited in the NCBI Gene Expression Omnibus (GEO) under the accession number GSE240971. The following datasets were also used: RRBS in E7.5 epiblast (GSM1471900), RRBS in sperm (GSM1471911), MethylC-Seq in male and female E13.5 PGCs (GSE56697), RNA-seq in *Dnmt1*-cKO E13.5 PGCs (GSE74938), RNA-seq in female *Setdb1*-cKO E13.5 PGCs (GSE60377), RNA-seq in female *Ezh2*-cKO E13.5 PGCs (GSE141182), H3K9me3 and H3K27me3 ChIP-seq in E13.5 PGCs (GSE60377). Source data are provided with this paper.

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

## Acknowledgements

This work was supported by the European Research Council (ERC Consolidator grant number 615371 to MW), the Agence Nationale de la Recherche (ANR-17-CE12-0013-02 to DB, ANR-23-CE12-0033 to MW), and the ITI InnoVec of the University of Strasbourg, CNRS, and Inserm (ANR-10-IDEX-0002 ANR-20-SFRI-0012 to MW). AB benefited from the support of the Fondation pour la Recherche Médicale. MM benefited from the support of the Région Grand Est. We thank Marina Peralta for advice on immunofluorescence quantification. We thank the IGBMC flow cytometry facility and the staff of the GenomEast sequencing platform for the RNA-sequencing experiments. GenomEast is a member of the 'France Genomique' consortium (ANR-10-INBS-0009).

## Author contributions

A.B. and M.W. designed and conceived the study. A.B. performed most experiments, including mouse breeding, isolation of samples, RRBS, molecular biology, and data analyses. M.M. assisted in the mouse work and dissections. M.D. performed the bioinformatic analyses of NGS data. M.K. and N.B.G. performed and interpreted histology experiments of the gonads. N.O. assisted in the immunofluorescence experiments. M.V.C.G. and D.B. performed and interpreted LUMA experiments. M.W.

supervised the study and performed data analyses. A.B. and M.W. wrote the manuscript with contributions from the other co-authors.

## Competing interests

The authors declare no competing interests.
