## [Transparent Peer Review file · Nature Communications]

UHRF2 mediates resistance to DNA methylation reprogramming in primordial germ cells

Corresponding Author: Dr Michael Weber

Version 0:

Reviewer comments:

Reviewer #1

(Remarks to the Author)

Mammalian primordial germ cells (PGCs) undergo global DNA demethylation, but yet the germline genes and evolutionarily young retrotransposons retain certain levels of methylation. However, the mechanism of persistent DNA methylation in PGCs has remained enigmatic. Bender et al. now show that UHRF2, a paralogue of well-characterized DNA methylation maintenance factor UHRF1, plays a key role in the resistance to demethylation. I believe that this is a breakthrough discovery in the field. On the whole, the study is well conducted, with the data well presented and the manuscript well written. Below are some comments for improvement.

1. The somatic phenotype of *Uhrf2*^{-/-} mice (Tm1b allele) is severer than that of *Uhrf2L1/L1* mice (Tm1d allele) and other *Uhrf2* mutant mice reported previously (Chen et al. 2017; Liu et al. 2017) (lines 182-190). The authors discuss that these differences could be attributed to the insertion cassette in *Uhrf2*^{-/-} animals or to the incomplete inactivation of all *Uhrf2* isoforms in other KO models (lines 390-392). These questions can be at least partly answered based on transcriptome data, such as the one shown in Fig. 3A, and it is not difficult to perform a similar analysis on *Uhrf2L1/L1* embryos. Are there any transcripts mapping to the insertion cassette in *Uhrf2*^{-/-} embryos? Do *Uhrf2L1/L1* embryos express downstream exons? Which *Uhrf2* isoforms are affected in these mutant embryos? These analyses should give valuable insights into the molecular basis of the phenotypes.
2. Considering that UHRF1 is a methylation maintenance factor, the presence of hyper-DMRs in *Uhrf2*^{-/-} and *Uhrf2L1/L1* spermatozoa is puzzling but interesting (lines 292-307). Can the authors discuss how this arises upon a lack of UHRF2?
3. While the importance of UHRF2 in methylation maintenance in PGCs is convincing, it is not clear how this factor, but not UHRF1, can maintain methylation at specific sequences. Can the authors discuss the mechanism contributing to the selectivity?

Minor points

1. Fig. 1B: The panel should contain an appropriate control (non-germline genes) if the authors want to use this panel to discuss delayed demethylation of the germline genes (lines 127-129). Or, perhaps Fig. 1C alone may be enough?
2. Fig. 1D: Residually methylated regions (RMRs) are defined by the presence of at least 4 CpGs with 30% methylation (lines 133-136). Fig. 1D is a bit confusing because it looks as if many RMRs have less than 30% methylation. I believe that the graphs show the methylation levels of individual CpGs within the RMRs. Please make it clear (perhaps in the legend?).
3. Lines 144-148: I am not sure about the statement that the temporal data obtained by reduced representation bisulfite sequencing (RRBS) "strongly suggests" a maintenance mechanism rather than de novo activity. I think that the temporal pattern is only "consistent with" the maintenance mechanism.
4. Fig. 2B: The panel shows that *Uhrf2* expression is not downregulated in PGCs (lines 171-172), but the readers may be interested in the actual levels of *Uhrf1* and *Uhrf2* transcripts. Addition of a graph (as a Supplementary Figure?) showing the actual FPKM or TPM values will be helpful.

Reviewer #2

(Remarks to the Author)

In this report, Bender and colleagues investigated the role of *Uhrf2* in primordial germ cell development. They hypothesized that UHRF2 [a paralogue of UHRF1, which in somatic cells complexes with DNMT1 to maintain DNA methylation during

DNA replication] may contribute to the resistance of certain genomic elements to become demethylated during PGC development.

The authors profiled the DNA methylation status at various time points of PGC development by using the RRBS methodology and confirmed and extended published results. They incorporated public datasets to their analysis to define “residually methylated regions” (RMRs) that exhibit higher DNA methylation at E13.5 compared to the globally lowly methylated genome.

Next, the authors generated and characterized a constitutive and a conditional deficiency mouse model, depleting Uhrf2 from PGCs. They discovered that DNA demethylation refractory RMR regions, enriched in repetitive elements, were fully hypomethylated in Uhrf2 depleted PGCs. Strikingly, DNA hypomethylation did not impact on transcription of various endogenous repetitive elements that had lost their DNA methylation. Nevertheless, while male PGCs transcriptomes were grossly unaltered, a plethora of meiosis related genes were upregulated in mutant female PGCs. In accordance, the authors observed a major reduction in the number of oocytes formed from Uhrf2 depleted PGCs which resulted in female specific infertility. In contrast, paternal gametes generated from Uhrf2 depleted PGCs retained their fertilization potential even though these cells displayed changes in DNA methylation levels at RMRs as well as at other sites, the later presumably resulting from indirect regulatory effects.

The authors investigated in this paper the role of Uhrf2 in DNA methylation reprogramming processes occurring during PGC development. They resolve an important long-standing mechanistic question which is relevant to a wider range of scientists interested in epigenetic reprogramming as well as in germ cell development. The experiments and the computational analysis of this study have been well designed and performed. The paper is easy to read. A few points remain to be addressed before the paper is suitable for publication.

- 1) As RRBS covers the genomic CpGs only partially, it is important to report in Figure 1A the number of CpGs in each context (CGI vs non-CGI) that are presented in the data.
- 2) Figures 1B and 1C: please include CG methylation for male PGCs for E12.5-E14.5.
- 3) According to the Methods section, the RRBS data mapping was not conducted at the consensus sequence of each repeat family, but rather at the whole genome mm10. Subsequently, the mapped CpG sites were overlapped with the UCSC Repeat Masker database, which contains genomic coordinates of individual copies of each repeat family. Are the numbers of individual copies in a given family presented in Figure 1G equal between the WGBS and RRBS datasets? It is presumed that RRBS covers fewer copies compared to WGBS, but the figure does not clearly indicate whether commonly covered repeats are presented or not.
- 4) Further, it would be informative to report either (a) the number of repeat members covered from each family (overlapping with a 500bp tile which meets the criteria of a minimum of 4 sequenced CpGs) and the percentage of the covered repeat copies relative to the total number of copies in each family, or (b) the number of RRBS covered CpGs overlapping with a given repeat family, and the percentage relative to the total number of CpGs in that family.
- 5) Finally, considering potential variability in DNA methylation levels, instead of reporting the mean percentage methylation for each family, it would be more informative to present the data in boxplots or violin plots depicting either the DNA methylation at each copy of the examined family or the distribution of methylated CpG percentages that overlap with each family. This approach aligns with the data presentation for RMR methylation in Figure 1J. It is recommended to update according to this comment the Figures 1G, 3F and 5J.
- 6) Figure 1D: As the RMRs were defined by WGBS, what is the fraction of them covered in RRBS?
- 7) Figure 2C: how is UHRF1 expressed in Uhrf2 mutant PGCs? Please include IF data for UHRF1 in wt and Uhrf2 mutant cells.
- 8) Figure 2E: where is the epitope of the antibody for UHRF2 localized in the protein? In the lacZ model, is there a chimeric protein formed? Vice versa, is there mRNA and protein expression of from the conditional deficient allele?
- 9) Figure 2G-2I: why is there a weight difference between the null mice of the two models? Does the growth retardation and lethality run in specific breeding pairs?
- 10) Figure 2I: Indicate the genetic crosses in legend.
- 11) Figure 3C: is there a difference between the sexes in the reduction of the number of PGCs at E13.5? Are PGCs in females more affected? To what extent is the flow cytometry read-out influenced by the change in transgenic reporter expression? Are the PGCs dying or prematurely differentiating?
- 12) Figure 4D: what is the reason for downregulation of pluripotency genes in Uhrf2 mutant PGCs? Is this a direct effect or resulting from precocious entry into meiosis? Do mutant male PGC demonstrate a similar decrease in expression of pluripotency markers?
- 13) Figure 4G. Dnmt1 cko PGCs: why is there “only” a limited (yet significant) overlap of genes upregulated between the two genotypes? Does the limited overlap reflect differences in the collection of cells (or their developmental timing) or in biology?
- 14) Figure 5B: Please show data for primary, secondary and antral follicles separately.
- 15) Figure S5C: the data in this figure shows that some meiotic genes are upregulated even in male mutant PGCs, yet possibly non-significant. How do the authors interpret this notion? This should be discussed in the manuscript.
- 16) Figure 5I: Concerning the meiosis-related genes, upregulated in mutant female PGCs, are there any sequence motifs enriched in the promoters of such response genes, such as of Dmrt1 or Meiosin?
- 17) Figure 5I: What are the sequence/genomic features of the hyper DMRs? Where are they located in the genome? Are they linked to transcriptional units? Could they function as enhancers?
- 18) Do female mutant PGCs enter meiosis precociously?
- 19) Finally, to be able to integrate these new findings with knowledge of existing literature (as discussed in the discussion of the paper), it is important to integrate epigenomic (H3K9me3, H3K27me3) and transcriptional responses measured in wildtype cells and in various mutants (Setdb1, Ezh2/PRC2, Rnf2/PRC1) from other studies into the current analysis. In other words, which genes and repetitive elements controlled by Uhrf2 are also controlled by these other chromatin modifiers? What detailed molecular insights can be obtained with respect to the hierarchy and interplay between the different regulators (see e.g. Yokobayashi et al., Nature, 2012; Liu et al., Genes Dev, 2014; Huang, Nature, 2021)?

Reviewer #3

(Remarks to the Author)

In this manuscript, entitled “UHRF2 mediates resistance to DNA methylation (DNAm) reprogramming in primordial germ cells”, Bender and colleagues first identify “residually methylated regions” (RMRs) in the male and female early germline, and identify specific transposable element (TE) families as enriched within these regions. Subsequently, they study the expression profile of UHRF2, the enigmatic paralog of the DNMT1 cofactor UHRF1, in the germline and describe a UHRF2 knock-out mouse and its phenotype. They then show via RRBS that UHRF2 is required for resistance to DNA demethylation in PGCs- ie at RMRs; while *Uhrf2* knock-out mice show no change in DNAm in somatic cells, PGCs show clear loss of DNAm specifically at retrotransposons that otherwise retain this epigenetic mark. Such loss of DNAm is not associated with changes in the expression of such retrotransposons, revealing that other mechanisms must compensate for retrotransposon control in these germ cells. Notably, precocious demethylation of specific germline genes is also observed in *Uhrf2*-deficient PGCs, and specific meiotic genes are overexpressed in females but not males. Overexpression of such genes is accompanied by impaired oocyte development and infertility. *Uhrf2* loss also leads to incomplete remethylation of retrotransposons during spermatogenesis. Taken together, these findings reveal a critical role for UHRF2 in controlling DNA methylation specifically in the germline.

This is a well written manuscript and the data is clearly presented in the figures. The overall story clearly novel and I believe appropriate for publication in Nature Communications, with minor revisions. Detailed comments/suggestions below.

Major points/questions

Results presented in Figure 4 reveal clearly that TEs remain largely repressed despite the loss of DNAm. Given that the authors already have the data, how does this TE expression result compare to DNMT1 KO male and female PGCs at the same time point?

Also in Figure 4 (panel G), while there is some overlap with genes upregulated in the DNMT1 KO PGCs. The majority of those de-repressed in the UHRF2 KO are not scored as upregulated in DNMT1 KO PGCs. The authors should at least comment on these genes. Are they even methylated in WT PGCs at the same time point? Ie are these indirect effects of the UHRF2 KO? Are they also enriched for meiosis or germline genes?

For the discussion:

Some meiotic genes are regulated by PRC1.6, as previously reported by the author, and a subset of these are also regulated by the K9 KMTase SETDB1, along with TEs. Given this, are the meiotic genes showing increased/premature expression (coincident with reduced/premature loss of DNAm) in the UHRF2^{-/-} female germline also upregulated prematurely in the absence of PRC1.6 subunits or SETDB1, if this has been studied in PGCs? Would support the connection to H3K9me3 if so.

Might the TEs showing reduced DNAm up through the spz stage overlap with those whose de novo DNAm depends on the piRNA pathway? The authors could compare families of TEs, or even specific elements in the genome (if mappability is not an issue) showing lower DNAm in late male germ cells, as reported in various published piRNA biogenesis factor mutants, with the TE families showing reduced DNAm in their UHRF2 KO. This would suggest that UHRF2 is required for maintenance of DNAm following piRNA-dependent de novo DNAm.

In the final paragraph, perhaps worth mentioning that in the early embryo/ICM DNAm levels are globally reduced, and as in PGCs, the residual DNAm is present at young TEs and UHRF1 is sequestered in the cytoplasm. Perhaps UHRF2 plays a role in maintenance of DNAm at these elements at this stage as well.

Minor points

LINE 92

“Here, we provide a comprehensive profiling of DNA methylation across normal PGC development (from embryonic stages E8.5 to E17.5)...” As RRBS was employed here, rather than WGBS, I do not think it is appropriate to refer to this valuable profiling series as “comprehensive”, given that most of the CpGs in the genome are not surveyed.

Figure 1E. Define RMR in the legend. What is there size for example?

Figure 1I. Separate female data from day 11.5 with a space, as done for the male data. Also, would be nice if this panel included the behavior/% CG meth of CGs not in the context of RMRs, ie to show what the baseline level of DNAm is in the genome at these time points.

Figure 1J legend. Add time point.

Figure 1K. Why not show the TE loci already shown in 1H here? ie the MuLV and/or IAPEz representative RMRs, rather than 2 different TEs?

Figure 4. RNA Harvest day for RNAseq analysis is not presented for several of the panels in this figure, nor is it mentioned in the figure legend. These should be added.

Version 1:

Reviewer comments:

Reviewer #1

(Remarks to the Author)

The authors responded properly to most of my criticisms except major point 3.

In this work, the authors argue that downregulation and sequestration of UHRF1 cause global DNA demethylation in PGCs but UHRF2 maintains methylation at specific sequences. This would suggest that UHRF1 is a general and fundamental maintenance factor but UHRF2 acts only on specific targets. In major point 3, I asked the authors to discuss how this functional difference arises in the two closely related proteins. While the authors discuss the role of the TTD and PHD domains in the Discussion (lines 359-362), this does not explain the difference because these domains are shared by both proteins. I believe that this is an important point and would like deeper discussions.

If the authors respond to this request properly, the paper is appropriate for publication.

Reviewer #2

(Remarks to the Author)

I have carefully reviewed the revised manuscript and appreciate the detailed and thoughtful responses provided to my comments and those of the other reviewers. The revisions have significantly enhanced the quality of the manuscript. In particular, the clarifications regarding the DNA methylation analysis (WGBS vs RRBS), the inclusion of additional data on the precocious expression of meiotic genes, and the integrative analysis of public datasets to further characterize the chromatin status of RMRs are noteworthy improvements.

I find the revisions satisfactory, and I have no further major comments or concerns. Therefore, I am pleased to recommend its acceptance for publication in its current form.

Reviewer #3

(Remarks to the Author)

The authors have addressed each of my concerns/points and modified the manuscript accordingly. I believe that this study is now ready for publication, adding a very interesting new twist to the regulation of DNA methylation in the germline.

We thank the three reviewers for their positive comments and constructive criticisms that contributed to improve the manuscript.

Reviewer #1 (Remarks to the Author):

Mammalian primordial germ cells (PGCs) undergo global DNA demethylation, but yet the germline genes and evolutionarily young retrotransposons retain certain levels of methylation. However, the mechanism of persistent DNA methylation in PGCs has remained enigmatic. Bender et al. now show that UHRF2, a paralogue of well-characterized DNA methylation maintenance factor UHRF1, plays a key role in the resistance to demethylation. I believe that this is a breakthrough discovery in the field. On the whole, the study is well conducted, with the data well presented and the manuscript well written. Below are some comments for improvement.

1. The somatic phenotype of *Uhrf2*^{-/-} mice (Tm1b allele) is severer than that of *Uhrf2*^{L1/L1} mice (Tm1d allele) and other *Uhrf2* mutant mice reported previously (Chen et al. 2017; Liu et al. 2017) (lines 182-190). The authors discuss that these differences could be attributed to the insertion cassette in *Uhrf2*^{-/-} animals or to the incomplete inactivation of all *Uhrf2* isoforms in other KO models (lines 390-392). These questions can be at least partly answered based on transcriptome data, such as the one shown in Fig. 3A, and it is not difficult to perform a similar analysis on *Uhrf2*^{L1/L1} embryos. Are there any transcripts mapping to the insertion cassette in *Uhrf2*^{-/-} embryos? Do *Uhrf2*^{L1/L1} embryos express downstream exons? Which *Uhrf2* isoforms are affected in these mutant embryos? These analyses should give valuable insights into the molecular basis of the phenotypes.

As requested by the reviewer, we performed RNA-seq in *Uhrf2*^{L1/L1} embryos to better characterize this mutant allele. We confirmed that this allele lacks the exon 3 but still expresses the downstream exons of the gene (see the updated **Supplementary Figure S3a**). This substantiates the possibility that the differences in phenotype could be caused by mRNA isoforms lacking the exon 3 that are still produced in *Uhrf2*^{L1/L1} but not *Uhrf2*^{-/-} animals and maybe produce truncated proteins from a downstream ATG. However, we have no further evidence to support this hypothesis because the UHRF2 antibody used in this study cannot detect truncated isoforms because it recognizes an epitope encoded by the exon 3 of the gene. We now discuss this hypothesis in more details in **the discussion section page 12**:

*“It is conceivable that naturally occurring mRNA isoforms lacking the exon 3, which are disrupted in the *Uhrf2*^{-/-} but not in the *Uhrf2*^{L1/L1} model, could produce truncated proteins from a downstream ATG. Such truncated proteins cannot be detected by the antibody used in this study because it recognizes an epitope encoded by the exon 3 of the gene.”*

2. Considering that UHRF1 is a methylation maintenance factor, the presence of hyper-DMRs in *Uhrf2*^{-/-} and *Uhrf2*^{L1/L1} spermatozoa is puzzling but interesting (lines 292-307). Can the authors discuss how this arises upon a lack of UHRF2?

We agree that the appearance of hyper-DMRs is puzzling considering the potential maintenance function of UHRF2. As requested by the reviewer 2, we analyzed the genomic features of hyper-DMRs and did not find an enrichment in specific genome compartments (new **Supplementary Fig S10f**). We included the following text in the discussion to speculate on how this could arise upon lack of UHRF2:

“Moreover, the discovery of hyperDMRS in spermatozoa is very puzzling considering the potential maintenance function of UHRF2. It is possible that the chromatin landscape and recruitment of de novo methylation machinery are modified at some sequences in absence of UHRF2, or that the UHRF1/DNMT1 axis is stimulated in absence of UHRF2 leading to excessive maintenance methylation at some sequences during spermatogenesis.”

3. While the importance of UHRF2 in methylation maintenance in PGCs is convincing, it is not clear how this factor, but not UHRF1, can maintain methylation at specific sequences. Can the authors discuss the mechanism contributing to the selectivity?

We discuss in the introduction and discussion of the manuscript that UHRF1 is known from the literature to be absent from the nucleus of PGCs, which is the most likely explanation for why UHRF2, but not UHRF1, can maintain methylation during epigenetic reprogramming in PGCs. To further highlight this point, we included immunofluorescence data of UHRF1 in gonads, which confirmed that UHRF1 is abundantly detected in surrounding cells but undetectable in PGCs. These results are presented in the **new Supplementary Figure S5**.

Minor points

1. Fig. 1B: The panel should contain an appropriate control (non-germline genes) if the authors want to use this panel to discuss delayed demethylation of the germline genes (lines 127-129). Or, perhaps Fig. 1C alone may be enough?

Following the reviewer's suggestion, we added a control showing early demethylation (in the non-germline gene *Etv6*) in the **Figure 1b**.

2. Fig. 1D: Residually methylated regions (RMRs) are defined by the presence of at least 4 CpGs with 30% methylation (lines 133-136). Fig. 1D is a bit confusing because it looks as if many RMRs have less than 30% methylation. I believe that the graphs show the methylation levels of individual CpGs within the RMRs. Please make it clear (perhaps in the legend?).

Yes, **Figure 1d** shows the methylation of individual CpGs within RMRs. We chose this representation because it allows for a better comparison with the CpGs in the whole genome and analysis of RRBS data, but agree that it can be a bit confusing. This has now been clarified in the legend of **Figure 1d**, but also **Figure 1j** and **Figure 3e**.

3. Lines 144-148: I am not sure about the statement that the temporal data obtained by reduced representation bisulfite sequencing (RRBS) “strongly suggests” a maintenance mechanism rather than de novo activity. I think that the temporal pattern is only “consistent with” the maintenance mechanism.

We agree with the reviewer and replaced “strongly suggesting” with “which is consistent with” in the text.

4. Fig. 2B: The panel shows that *Uhrf2* expression is not downregulated in PGCs (lines 171-172), but the readers may be interested in the actual levels of *Uhrf1* and *Uhrf2* transcripts. Addition of a graph (as a Supplementary Figure?) showing the actual FPKM or TPM values will be helpful.

We now provide both normalized and raw FPKM values in the source data file linked to the Figure 2b. We also added this sentence in the legend of **Figure 2b**: “Raw FPKM values are provided in the source data file”.

Reviewer #2 (Remarks to the Author):

In this report, Bender and colleagues investigated the role of Uhrf2 in primordial germ cell development. They hypothesized that UHRF2 [a paralogue of UHRF1, which in somatic cells complexes with DNMT1 to maintain DNA methylation during DNA replication] may contribute to the resistance of certain genomic elements to become demethylated during PGC development.

The authors profiled the DNA methylation status at various time points of PGC development by using the RRBS methodology and confirmed and extended published results. They incorporated public datasets to their analysis to define “residually methylated regions” (RMRs) that exhibit higher DNA methylation at E13.5 compared to the globally lowly methylated genome.

Next, the authors generated and characterized a constitutive and a conditional deficiency mouse model, depleting Uhrf2 from PGCs. They discovered that DNA demethylation refractory RMR regions, enriched in repetitive elements, were fully hypomethylated in Uhrf2 depleted PGCs. Strikingly, DNA hypomethylation did not impact on transcription of various endogenous repetitive elements that had lost their DNA methylation. Nevertheless, while male PGCs transcriptomes were grossly unaltered, a plethora of meiosis related genes were upregulated in mutant female PGCs. In accordance, the authors observed a major reduction in the number of oocytes formed from Uhrf2 depleted PGCs which resulted in female specific infertility. In contrast, paternal gametes generated from Uhrf2 depleted PGCs retained their fertilization potential even though these cells displayed changes in DNA methylation levels at RMRs as well as at other sites, the later presumably resulting from indirect regulatory effects.

The authors investigated in this paper the role of Uhrf2 in DNA methylation reprogramming processes occurring during PGC development. They resolve an important long-standing mechanistic question which is relevant to a wider range of scientists interested in epigenetic reprogramming as well as in germ cell development. The experiments and the computational analysis of this study have been well designed and performed. The paper is easy to read. A few points remain to be addressed before the paper is suitable for publication.

We thank this reviewer for making pertinent suggestions and raising many interesting points that we have addressed in the revised manuscript.

1) As RRBS covers the genomic CpGs only partially, it is important to report in Figure 1A the number of CpGs in each context (CGI vs non-CGI) that are presented in the data.

We agree with the reviewer. We now provide the number of covered CpGs in CGI and non-CGI for each datapoint in the source data file and added the following sentence in the legend of **Figure 1a**:

“The average number of covered CpGs per sample is $n=599050$ for CGI and $n=560437$ for non-CGI”.

2) Figures 1B and 1C: please include CG methylation for male PGCs for E12.5-E14.5.

As requested by the reviewer, we included CG methylation values for male PGCs (E12.5-14.5) in the **Figures 1b and 1c**.

3) According to the Methods section, the RRBS data mapping was not conducted at the consensus sequence of each repeat family, but rather at the whole genome mm10. Subsequently, the mapped CpG sites were overlapped with the UCSC Repeat Masker database, which contains genomic coordinates of individual copies of each repeat family. Are the numbers of individual copies in a given family presented in Figure 1G equal between the WGBS and RRBS datasets? It is presumed that RRBS covers fewer copies compared to WGBS, but the figure does not clearly indicate whether commonly covered repeats are presented or not.

4) Further, it would be informative to report either (a) the number of repeat members covered from each family (overlapping with a 500bp tile which meets the criteria of a minimum of 4 sequenced CpGs) and the percentage of the covered repeat copies relative to the total number of copies in each family, or (b) the number of RRBS covered CpGs overlapping with a given repeat family, and the percentage relative to the total number of CpGs in that family.

Response to the points 3) and 4) above.

We thank the reviewer for raising these important points. **Figure 1g** shows the methylation of CpGs in all individual copies covered in each dataset and not only the common ones between WGBS and RRBS. We apologize if this was not clear. Our intention was to show that RRBS provides a fairly good estimate of TE methylation compared to WGBS. Nevertheless, the reviewer is right that RRBS covers fewer individual copies in each family compared to WGBS. We have now quantified this in detail by calculating the number of individual copies covered in each dataset and the percentage of covered copies relative to the total number of copies for each family. We report that WGBS covers on average 69% of the total number of copies and RRBS covers on average 12% of the total number of copies. We provide all the numbers in the source data file and modified the legend of **Figure 1g** to clarify this point:

“The boxplots show the distribution of methylation levels of individual CpGs overlapping with individual copies of each retrotransposon family in the WGBS or RRBS datasets. On average, 69% and 12% of the total number of individual genomic copies are covered in the WGBS and RRBS datasets, respectively. The numbers of CpGs and individual copies covered in each dataset are given in the source data file.”

5) Finally, considering potential variability in DNA methylation levels, instead of reporting the mean percentage methylation for each family, it would be more informative to present the data in boxplots or violin plots depicting either the DNA methylation at each copy of the examined family or the distribution of methylated CpG percentages that overlap with each family. This approach aligns with the data presentation for RMR methylation in Figure 1J. It is recommended to update according to this comment the Figures 1G, 3F and 5J.

We agree with the reviewer and replaced the initial barplots of mean methylation with boxplots depicting the distribution of methylation scores for individual CpGs overlapping with each TE family in the **Figure 1g**. This now aligns with the representation of RMR methylation in the Figures 1d and 1j. If the reviewer agrees, we prefer to keep the current representation for the **Figures 3f and 5j** because we have several biological replicates from independent animals (up to n=4) and the barplots of mean methylation allow us to show the biological variability between replicates in reasonably sized figures.

6) Figure 1D: As the RMRs were defined by WGBS, what is the fraction of them covered in RRBS?

Our set of RMRs contains 350,807 CpGs. We quantified that an average of $32,473 \pm 1,271$ CpGs ($9.26 \pm 0.36\%$) were covered in the various RRBS datasets generated in mutant PGCs. To clarify this point, we added the following text in the method section:

“We then selected all CpGs common to RMRs identified in male and female E13.5 PGCs (n= 350,807 CpGs) for further analysis. From this set, an average of $32,473 \pm 1,271$ CpGs ($9.26 \pm 0.36\%$) were covered in the various RRBS datasets generated in mutant PGCs.”

7) Figure 2C: how is UHRF1 expressed in Uhrf2 mutant PGCs? Please include IF data for UHRF1 in wt and Uhrf2 mutant cells.

As requested by the reviewer, we performed IF of UHRF1 in WT and Uhrf2 mutant gonads at E13.5. The results confirmed that UHRF1 protein is abundantly detected in surrounding somatic cells but not PGCs of WT embryos, and that there is no compensatory expression of UHRF1 protein in PGCs of Uhrf2 mutant embryos. These results are presented in the **new Supplementary Figure S5** and we added the following text page 6:

“First, we examined the expression of UHRF1 by immunostaining in WT and Uhrf2 mutant gonads at E13.5. Similar to the WT condition, UHRF1 was detected in the surface epithelium and in some interstitial cells of the ovaries and testes but remained undetectable in PGCs of Uhrf2 knockout gonads (Supplementary Fig S5), indicating that there is no compensatory ectopic expression of UHRF1 upon loss of UHRF2 in PGCs.”

8) Figure 2E: where is the epitope of the antibody for UHRF2 localized in the protein? In the lacZ model, is there a chimeric protein formed?

The UHRF2 C-10 monoclonal antibody (Santa Cruz sc-398953) was raised against amino acids 144-208 of the human UHRF2 protein, which corresponds to the region encoded by the exon 3 in the mouse gene. Therefore, this antibody cannot detect whether chimeric or truncated proteins are produced from mutant alleles because both our genetic models disrupt the exon 3.

Vice versa, is there mRNA and protein expression of from the conditional deficient allele?

To determine if mRNAs are produced from the conditional deficient allele, we generated RNA-seq in *Uhrf2^{L1/L1}* embryos (updated **Figure S3a**). We confirmed that this mutant allele produces full length mRNAs lacking the exon 3 but at reduced levels probably due to NMD (**Figure S3a**).

9) Figure 2G-2I: why is there a weight difference between the null mice of the two models? Does the growth retardation and lethality run in specific breeding pairs?

We share the reviewer's interest in these questions but unfortunately have no definitive answer at this point. The growth retardation in *Uhrf2*^{-/-} animal occurs randomly and is not restricted to specific breeding pairs. One potential explanation could be indirect effects caused by the insertion cassette in the *Uhrf2*^{-/-} model. One alternative hypothesis could involve the production of truncated proteins from a downstream ATG in mRNAs lacking the exon 3 that naturally occur in embryos. Indeed, these mRNAs are still expressed in *Uhrf2^{L1/L1}* but not *Uhrf2*^{-/-} animals that lack all transcription beyond the exon 3,

which could contribute to the different phenotypes. As discussed above, such truncated proteins could unfortunately not be detected by our antibody, which precludes us from investigating this hypothesis further. We have amended **the discussion page 12** to discuss this possibility in more detail:

“For example, it is conceivable that naturally occurring mRNA isoforms lacking the exon 3, which are disrupted in the $Uhrf2^{-/-}$ but not in the $Uhrf2^{L1/L1}$ model, could produce truncated proteins from a downstream ATG. Such truncated proteins cannot be detected by the antibody used in this study because it recognizes an epitope encoded by the exon 3 of the gene.”

10) Figure 2I: Indicate the genetic crosses in legend.

We now indicate the genetic crosses in the legend of **Figure 2i**.

11) Figure 3C: is there a difference between the sexes in the reduction of the number of PGCs at E13.5? Are PGCs in females more affected?

We observed that both sexes show a similar reduction of the number of PGCs in E13.5 mutant embryos. To illustrate this, we revised the **Figure 3c** to include the sex information in the data.

To what extent is the flow cytometry read-out influenced by the change in transgenic reporter expression?

This is a valid concern but our data suggest that the flow cytometry results are not influenced by changes in transgenic reporter expression. Indeed, we observed reduced numbers of male PGCs even though male PGCs express normal levels of Oct4 in mutant embryos (see below and **revised Figure 4d**), suggesting that changes in PGC numbers are not caused by changes in Oct4 reporter expression.

Are the PGCs dying or prematurely differentiating?

To address this question, we investigated apoptosis by TUNEL assay in E13.5 gonads and found no sign of apoptosis in mutant E13.5 PGCs (data not shown). This indicates that the reduced number of PGCs is not explained by cell death at E13.5. In contrast, we now provide evidence that mutant female PGCs enter meiosis prematurely (see detailed response below).

12) Figure 4D: what is the reason for downregulation of pluripotency genes in $Uhrf2$ mutant PGCs? Is this a direct effect or resulting from precocious entry into meiosis? Do mutant male PGC demonstrate a similar decrease in expression of pluripotency markers?

We note that previous related studies similarly found a downregulation of pluripotency genes in female PGCs upon inactivation of DNMT1 (Hargan-Calvopina et al. 2016), PRC1 (Yokobayashi et al. 2013) or EED (Lowe et al. 2022), and believe that it indicates precocious entrance into meiosis (see below). We revised the **Figure 4d** to include data from male PGCs and show that male PGCs do not demonstrate a similar decrease in the expression of pluripotency markers, further suggesting that the downregulation of pluripotency genes is linked to entry into meiosis that is observed only in female PGCs.

13) Figure 4G. $Dnmt1$ cko PGCs: why is there “only” a limited (yet significant) overlap of genes upregulated between the two genotypes? Does the limited overlap reflect differences in the collection of cells (or their developmental timing) or in biology?

We agree that the overlap in **Figure 4g** is not complete, but consider it far from “limited” and highly significant ($p=9.94e-81$). Several factors can explain imperfect overlaps in RNA-seq datasets:

1. Biological and technical noise.
2. Differences in experimental settings. One factor could be the mouse genetic background, which is not specified in the *Dnmt1* cKO study. Furthermore, although PGCs are collected at the same developmental time point (E13.5), the *Dnmt1* cKO study uses a different Oct4 reporter mouse strain (Oct4-IRES-Gfp).
3. The threshold effect of Venn diagrams. Venn diagrams rely on arbitrary thresholds and tend to underrepresent overlaps when many genes are just under the threshold in one dataset. In the case of **Figure 4g**, the *Dnmt1* cKO RNA-seq dataset has less statistical power and therefore many genes upregulated in *Uhrf2*^{-/-} PGCs are just above the significance threshold ($p_{adj}<0.01$) in the *Dnmt1* cKO dataset, as exemplified by the list of meiotic genes below:

Gene	Uhrf2 ^{-/-} PGCs		Dnmt1 cKO PGCs	
	Log2(FC)	p _{adj}	Log2(FC)	p _{adj}
Tuba3b	1.87	1.0e-12	1.01	0.097
Zfp541	3.41	7.7e-12	2.55	0.045
Nxf2	3.11	3.4e-4	2.58	0.012
Ccnb3	2.68	3.2e-12	2.63	0.019
Sycp3	1.70	9.5e-11	2.43	0.019
Btbd18	1.35	7.0e-10	1.59	0.031
Slc25a31	1.20	5.1e-7	1.39	0.024
Tdrd9	1.74	1.7e-6	1.87	0.038
M1ap	1.04	3.1e-5	2.62	0.010
Tex11	1.29	1.1e-6	1.31	0.014
Dmc1	1.34	7.1e-6	2.13	0.038

14) Figure 5B: Please show data for primary, secondary and antral follicles separately. Unfortunately, we did not collect this data.

15) Figure S5C: the data in this figure shows that some meiotic genes are upregulated even in male mutant PGCs, yet possibly non-significant. How do the authors interpret this notion? This should be discussed in the manuscript.

The reviewer is right that a small number of meiotic genes show a minor trend of upregulation in male mutant PGCs, yet this is mostly in the background signal and only two of them (*Sycp1* and *Tuba3b*) are significant. Detailed values of expression in male PGCs can be found in the **Supplementary Table S4**. We interpret this by the fact that mutant male PGCs also undergo precocious demethylation (see **Figure 4h**), which could unleash expression at some of these meiotic genes but at very low levels because male PGCs are still blocked for meiosis at this time point. We added the following sentence in the results section to discuss this point:

“Some meiotic genes also show a minor trend of upregulation in male PGCs but at very low levels and mostly below the significance threshold (Supplementary Figure S6c), which likely reflects that male PGCs are still blocked for meiosis at this stage.”

16) Figure 5I: Concerning the meiosis-related genes, upregulated in mutant female PGCs, are there any sequence motifs enriched in the promoters of such response genes, such as of *Dmrt1* or *Meiosin*? To address this point, we performed a motif analysis in the promoters of upregulated genes using TFmotifView. This analysis did not reveal biologically meaningful hits. The only significant top hit was

NRF1 (binding motif GCGCNTGCGC), which probably reflects the CG richness of the promoters of meiosis-related genes.

17) Figure 5I: What are the sequence/genomic features of the hyper DMRs? Where are they located in the genome? Are they linked to transcriptional units? Could they function as enhancers?

We analyzed the genomic features of the hyper-DMRs in spermatozoa. The analysis revealed that hyper-DMRs occur equally inside and outside of genes and at this stage, we were unable to find an enrichment of hyper-DMRs in specific sequence types or functions. The proportions of sequence features in the hyper-DMRs are presented in the **new Supplementary Fig S10f**.

18) Do female mutant PGCs enter meiosis precociously?

We thank again the reviewer for bringing up this pivotal point that was missing in our initial manuscript.

We analyzed meiosis by performing immunofluorescence of various meiotic markers (STRA8, REC8, SYCP3, γ H2AX) in female E13.5 gonads of mutant and control animals. While there were as expected no signs of meiosis in WT gonads, we detected PGCs positive for meiotic markers in mutant ovaries, confirming that mutant female PGCs enter meiosis prematurely. These new results are presented in the **Figure 4i** and the new **supplementary Figure S9**, and discussed in the **results section page 8**:

“The overexpression of key meiotic genes suggests that mutant female PGCs could enter meiosis precociously, which we investigated by immunostainings in E13.5 gonads. In Uhrf2 mutant E13.5 ovaries, several STRA8-positive cells were observed, indicating they are meiotic, whereas such STRA8-positive cells were rarely detected in WT littermates (Fig 4i). In agreement with the conclusion that germ cells had initiated meiosis at an earlier stage, many germ cells were REC8-, SYCP3- or γ H2AX-positive at E13.5 in Uhrf2 mutant ovaries, while they were scarce or absent in control ovaries (Supplementary Fig S9). In contrast, meiotic germ cells were never observed in Uhrf2-deficient testes at E13.5 (data not shown). These results confirm a role of UHRF2 in regulating meiotic initiation in female PGCs.”

19) Finally, to be able to integrate these new findings with knowledge of existing literature (as discussed in the discussion of the paper), it is important to integrate epigenomic (H3K9me3, H3K27me3) and transcriptional responses measured in wildtype cells and in various mutants (Setdb1, Ezh2/PRC2, Rnf2/PRC1) from other studies into the current analysis. In other words, which genes and repetitive elements controlled by Uhrf2 are also controlled by these other chromatin modifiers? What detailed molecular insights can be obtained with respect to the hierarchy and interplay between the different regulators (see e.g. Yokobayashi et al., Nature, 2012; Liu et al., Genes Dev, 2014; Huang, Nature, 2021)?

We thank the reviewer for these valuable suggestions.

First, we integrated Chip-seq (H3K9me3, H3K27me3) and transcriptional responses measured previously in PGCs mutant for *Setdb1* and *Ezh2* (Liu et al. Genes Dev 2014; Huang et al. Nature 2021) to determine if RMRs/TEs are coregulated by these other chromatin pathways. This confirmed that RMRs are marked by the histone marks H3K9me3 and H3K27me3 in E13.5 PGCs and overexpressed in PGCs mutant for *Setdb1* or *Ezh2*, which supports that these chromatin marks likely compensate for the absence of DNA

methylation in PGCs. These results are presented in the **new supplementary Fig S7** and we added the following text in **the manuscript page 7**:

*“Indeed, the reanalysis of published datasets (Huang et al., 2021; S. Liu et al., 2014) showed that RMRs are marked by the histone marks H3K9me3 and H3K27me3 in E13.5 PGCs (**Supplementary Fig S7b**) and that evolutionary young TE families including IAPs are overexpressed in PGCs mutant for *Setdb1* or *Ezh2* (**Supplementary Fig S7c-d**), which substantiates the idea that these chromatin marks compensate for the absence of DNA methylation in PGCs.”*

Second, we included a comparison of the genes upregulated in female *Uhrf2*^{-/-} PGCs with the genes found upregulated in previous studies of PGCs mutant for *Rnf2*, *Setdb1* and *Ezh2* (Yokobayashi et al. Nature 2013; Liu et al. Genes Dev 2014; Huang et al. Nature 2021). This reveals that many meiotic genes controlled by *Uhrf2* are also repressed by these other chromatin pathways indicating convergence of these different epigenetic pathways to regulate the timing of expression of meiotic genes in female PGCs. These results are presented in the **new supplementary Fig S8** and we added the following text in **the manuscript page 8**:

*“Interestingly, a significant proportion of meiotic genes upregulated by mutation of *Uhrf2* are commonly upregulated in female PGCs mutant for *Setdb1*, *Ezh2* or *Rnf2* (Huang et al., 2021; S. Liu et al., 2014; Yokobayashi et al., 2013) (**Supplementary Fig S8**), which indicates a convergence of different epigenetic pathways to regulate the timing of expression of meiotic genes in female PGCs.”*

Reviewer #3 (Remarks to the Author):

In this manuscript, entitled “UHRF2 mediates resistance to DNA methylation (DNAm) reprogramming in primordial germ cells”, Bender and colleagues first identify “residually methylated regions” (RMRs) in the male and female early germline, and identify specific transposable element (TE) families as enriched within these regions. Subsequently, they study the expression profile of UHRF2, the enigmatic paralog of the DNMT1 cofactor UHRF1, in the germline and describe a UHRF2 knock-out mouse and its phenotype. They then show via RRBS that UHRF2 is required for resistance to DNA demethylation in PGCs- ie at RMRs; while *Uhrf2* knock-out mice show no change in DNAm in somatic cells, PGCs show clear loss of DNAm specifically at retrotransposons that otherwise retain this epigenetic mark. Such loss of DNAm is not associated with changes in the expression of such retrotransposons, revealing that other mechanisms must compensate for retrotransposon control in these germ cells. Notably, precocious demethylation of specific germline genes is also observed in *Uhrf2*-deficient PGCs, and specific meiotic genes are overexpressed in females but not males. Overexpression of such genes is accompanied by impaired oocyte development and infertility. *Uhrf2* loss also leads to incomplete remethylation of retrotransposons during spermatogenesis. Taken together, these findings reveal a critical role for UHRF2 in controlling DNA methylation specifically in the germline.

This is a well written manuscript and the data is clearly presented in the figures. The overall story clearly novel and I believe appropriate for publication in Nature Communications, with minor revisions. Detailed comments/suggestions below.

Major points/questions

Results presented in Figure 4 reveal clearly that TEs remain largely repressed despite the loss of DNAm. Given that the authors already have the data, how does this TE expression result compare to DNMT1 KO male and female PGCs at the same time point?

We agree with the reviewer that it is valuable to compare the expression of TEs in PGCs mutant for *Uhrf2* and *Dnmt1*. To answer this question, we reanalyzed RNA-seq data from *Dnmt1* conditional KO E13.5 PGCs (Hargan-Calvopina et al., 2016) to precisely quantify the expression of TE families. Similarly to *Uhrf2* KO PGCs in **Figure 4a**, we found no dysregulation in male PGCs and very little perturbations in female PGCs, in particular no upregulation of most of the methylated TE families such as IAPs. This confirms that TEs remain largely repressed in both mutants despite the loss of DNA methylation. We present these data in the **new Supplementary Figure S7a** and added the following text in the **manuscript page 7**:

*“This is corroborated by the reanalysis of published RNA-seq data from *Dnmt1*-deficient PGCs (Hargan-Calvopina et al., 2016) showing very little transcriptional deregulation of methylated TE families (Supplementary Fig S7a).”*

Also in Figure 4 (panel G), while there is some overlap with genes upregulated in the DNMT1 KO PGCs. The majority of those de-repressed in the UHRF2 KO are not scored as upregulated in DNMT1 KO PGCs. The authors should at least comment on these genes. Are they even methylated in WT PGCs at the same time point? Is are these indirect effects of the UHRF2 KO? Are they also enriched for meiosis or germline genes?

We thank the reviewer for raising this valid point. The reviewer #2 asked a similar question as to why the overlap between the genes upregulated in *Uhrf2* KO PGCs and *Dnmt1* KO PGCs is not better (see question 13). Our answer is that overlaps in Venn diagrams are never perfect due to technical, biological or statistical variations. Related to the latter, Venn diagrams rely on arbitrary thresholds and tend to underrepresent overlaps when many genes are just under the threshold in one dataset. In the case of **Figure 4g**, the *Dnmt1* cKO RNA-seq dataset has less statistical power because of a lower number of replicates and therefore many genes upregulated in *Uhrf2*^{-/-} PGCs are just above the significance threshold ($p_{adj} < 0.01$) in the *Dnmt1* cKO dataset. This is the case for a high number of meiotic and germline genes as exemplified by the non-exhaustive list below:

Gene	Uhrf2 ^{-/-} PGCs		Dnmt1 cKO PGCs	
	Log2(FC)	p _{adj}	Log2(FC)	p _{adj}
Tuba3b	1.87	1.0e-12	1.01	0.097
Zfp541	3.41	7.7e-12	2.55	0.045
Nxf2	3.11	3.4e-4	2.58	0.012
Ccnb3	2.68	3.2e-12	2.63	0.019
Sycp3	1.70	9.5e-11	2.43	0.019
Btbd18	1.35	7.0e-10	1.59	0.031
Slc25a31	1.20	5.1e-7	1.39	0.024
Tdrd9	1.74	1.7e-6	1.87	0.038
M1ap	1.04	3.1e-5	2.62	0.010
Tex11	1.29	1.1e-6	1.31	0.014
Dmc1	1.34	7.1e-6	2.13	0.038

Nevertheless, we agree that some genes could be upregulated only in *Uhrf2* KO PGCs for many reasons and we added a sentence **in the text page 8** to comment on these possibilities:

“The genes upregulated only in Uhrf2 KO PGCs could reflect indirect effects or other functions of UHRF2 unrelated to DNA methylation.”

For the discussion:

Some meiotic genes are regulated by PRC1.6, as previously reported by the author, and a subset of these are also regulated by the K9 KMTase SETDB1, along with TEs. Given this, are the meiotic genes showing increased/premature expression (coincident with reduced/premature loss of DNAm) in the UHRF2^{-/-} female germline also upregulated prematurely in the absence of PRC1.6 subunits or SETDB1, if this has been studied in PGCs? Would support the connection to H3K9me3 if so.

We thank the reviewer for these questions. It would have been very interesting to compare with PGCs mutant for PRC1.6 subunits but, to our knowledge, this has not been studied. On the other hand, transcriptome analyses have been performed in PGCs mutant for *Setdb1*, but also mutant for the PRC1 subunit *Rnf2* and the PRC2 subunit *Ezh2* (Yokobayashi et al. Nature 2013; Liu et al. Genes Dev 2014; Huang et al. Nature 2021). We therefore performed a comparison of the genes upregulated in female *Uhrf2*^{-/-} PGCs with the genes found upregulated in these mutant PGCs. This revealed that many meiotic genes controlled by *Uhrf2* are also repressed by these other chromatin pathways, which supports a connection with H3K9me3 and polycomb to regulate the timing of expression of meiotic genes in female PGCs. These results are presented in the **new supplementary Fig S8** and we added the following text in **the manuscript page 8**:

“Interestingly, a significant proportion of meiotic genes upregulated by mutation of Uhrf2 are commonly upregulated in female PGCs mutant for Setdb1, Ezh2 or Rnf2 (Huang et al., 2021; S. Liu et al., 2014; Yokobayashi et al., 2013) (Supplementary Fig S8), which indicates convergence of different epigenetic pathways to regulate the timing of expression of meiotic genes in female PGCs.”

Might the TEs showing reduced DNAm up through the spz stage overlap with those whose de novo DNAm depends on the piRNA pathway? The authors could compare families of TEs, or even specific elements in the genome (if mappability is not an issue) showing lower DNAm in late male germ cells, as reported in various published piRNA biogenesis factor mutants, with the TE families showing reduced DNAm in their UHRF2 KO. This would suggest that UHRF2 is required for maintenance of DNAm following piRNA-dependent de novo DNAm.

We thank the reviewer for suggesting this interesting point. We reanalyzed WGBS datasets generated in *Mili*-KO (Molaro et al. Genes Dev 2014) and *Miwi2*-KO (Manakov et al. Cell Rep 2015) late male germ cells. We confirmed that *Mili*-KO has a more drastic impact on TE methylation than *Miwi2*-KO (as described by Manakov et al. Cell Rep 2015), and found that the TE families hypomethylated in *Uhrf2*-mutant spermatozoa are also hypomethylated in *Mili*-KO cells (see table below). However, we note some discrepancies, such as several TE families (*L1Md_Gf*, *ETnERV-int*, *MMERGLN-int*, *RLTR10-int*, *RLTR4_MM-int*) strongly hypomethylated in *Mili*-KO cells but only marginally hypomethylated in *Uhrf2*-KO cells (see table below). Furthermore, it is difficult to make strong conclusions about an overlap because we are comparing WGBS and RRBS datasets. In that context we propose to leave this possibility open in the discussion:

“Since methylation of retrotransposons is also reduced in late male germ cells mutant for piRNA biogenesis factors (Manakov et al., 2015; Molaro et al., 2014), it is possible that UHRF2 is required for maintenance of DNA methylation following piRNA-dependent de novo DNA methylation”.

	diff Uhrf2-KO	diff Uhrf2-L1L1	diff Mili-KO
RLTR45-int	-0.268244608	-0.259492623	-0.359009976
ERV4_2-I_MM-int	-0.184371046	-0.208248894	-0.233381829
RLTR1B-int	-0.183375452	-0.194947444	-0.324390087
IAPLTR2b	-0.174656924	-0.183319855	-0.376119943
IAPLTR2a2_Mm	-0.168036338	-0.165394451	-0.2237057
IAPLTR2a	-0.153359873	-0.153958802	-0.338488078
L1Md_F	-0.143228278	-0.140304249	-0.201202934
MURVY-int	-0.141156661	-0.109787597	-0.155379987
IAP-d-int	-0.129290032	-0.149756203	-0.174627259
IAPeY-int	-0.129079581	-0.109519048	-0.204947841
IAPLTR1a_Mm	-0.116264314	-0.133334637	-0.118207231
IAPeZ-int	-0.098902223	-0.094807813	-0.082121895
IAPLTR2_Mm	-0.098849828	-0.07553791	-0.261795068
L1Md_A	-0.094228373	-0.080822162	-0.309862224
L1Md_T	-0.08885009	-0.078996081	-0.282114453
MMETn-int	-0.088010644	-0.106889808	-0.336225216
L1Md_Gf	-0.077305079	-0.069744939	-0.337545014
ETnERV-int	-0.051937809	-0.066025417	-0.263741289
MMERGLN-int	-0.033828258	-0.048799106	-0.226726035
RLTR10-int	-0.026991649	-0.025665958	-0.391089112
RLTR4_MM-int	-0.007692518	-0.008364816	-0.286177817

Table: Differences in DNA methylation of TE families in late male germ cells mutant for Uhrf2 and Mili.

In the final paragraph, perhaps worth mentioning that in the early embryo/ICM DNAm levels are globally reduced, and as in PGCs, the residual DNAm is present at young TEs and UHRF1 is sequestered in the cytoplasm. Perhaps UHRF2 plays a role in maintenance of DNAm at these elements at this stage as well.

We agree and added the following text in the final paragraph of the **discussion page 13**:

“Perhaps UHRF2 could participate in the maintenance of DNA methylation at retrotransposons during the global reduction of genome methylation in preimplantation embryos, since UHRF1 is mainly localized in the cytoplasm at these stages (Maenohara et al., 2017).”

Minor points

LINE 92

“Here, we provide a comprehensive profiling of DNA methylation across normal PGC development (from embryonic stages E8.5 to E17.5)...” As RRBS was employed here, rather than WGBS, I do not think it is appropriate to refer to this valuable profiling series as “comprehensive”, given that most of the CpGs in the genome are not surveyed.

We agree with the reviewer and replaced "we provide a comprehensive profiling of DNA methylation" with "we profiled DNA methylation" in the text line 92.

Figure 1E. Define RMR in the legend. What is there size for example?

We now define RMRs in the legend of Figure 1 and added additional information on the average size of RMRs in the text **page 4**:

"These RMRs have an average size of 1626 bp in female E13.5 PGCs and 1724 bp in male E13.5 PGCs"

Figure 1I. Separate female data from day 11.5 with a space, as done for the male data. Also, would be nice if this panel included the behavior/% CG meth of CGs not in the context of RMRs, ie to show what the baseline level of DNAm is in the genome at these time points.

As requested, we updated the **Figure 1i** to separate the female data from the E11.5 data with a space. However, we could not add the % meth of all the CGs not in the context of RMRs because this figure shows RRBS data. Indeed, RRBS cannot be used to show a fair representation of the methylation of all the CGs in the genome because of its strong bias for CpG islands. One way to do it is to separate CGs in or outside of CpG islands like we did in the Figure 1a, but this would complicate the figure too much. We invite the reader to refer to the **Figure 1a** and **Figure 1g** for the baseline methylation at these time points.

Figure 1J legend. Add time point.

We added the time point (E13.5) in the axis label and legend of the **Figure 1j**.

Figure 1K. Why not show the TE loci already shown in 1H here? ie the MuLV and/or IAPEz representative RMRs, rather than 2 different TEs?

This is because **Figure 1h** and **Figure 1k** show WGBS and RRBS data respectively, and many regions covered in WGBS are not covered in RRBS. Indeed, the IAPEz element from **Figure 1h** is not covered in RRBS. Nevertheless, we now added RRBS data in the **Figure 1k** that cover part of the same MuLV-int element from **Figure 1h**.

Figure 4. RNA Harvest day for RNAseq analysis is not presented for several of the panels in this figure, nor is it mentioned in the figure legend. These should be added.

We thank the reviewer for pointing this out. We added the time point (E13.5) in the panels and legends of the **Figure 4**.

In this work, the authors argue that downregulation and sequestration of UHRF1 cause global DNA demethylation in PGCs but UHRF2 maintains methylation at specific sequences. This would suggest that UHRF1 is a general and fundamental maintenance factor but UHRF2 acts only on specific targets. In major point 3, I asked the authors to discuss how this functional difference arises in the two closely related proteins. While the authors discuss the role of the TTD and PHD domains in the Discussion (lines 359-362), this does not explain the difference because these domains are shared by both proteins. I believe that this is an important point and would like deeper discussions.

If the authors respond to this request properly, the paper is appropriate for publication.

We acknowledge that we missed to include a discussion of the mechanisms responsible for the difference in selectivity between UHRF2 (maintenance at specific TEs) and UHRF1 (maintenance genome-wide). We suggest that this is due to divergences in the domains that have been reported to modify the binding preferences with chromatin marks, or to different partners that could guide the selectivity of UHRF2. We modified the discussion as follows to address this point in more detail:

“UHRF1 and UHRF2 are related proteins sharing similar domains, which questions why UHRF2 only acts at transposable elements in PGCs whereas UHRF1 is a genome-wide maintenance factor. Transposable elements are preferentially targeted for the deposition of H3K9me3 in PGCs (Huang et al., 2021). The TTD and PHD domains are the least conserved domains between UHRF2 and UHRF1 (Fig 2a) and mediate higher specificity for H3K9me3 in UHRF2 compared to UHRF1 (Ginnard et al., 2022; Pichler et al., 2011; Vaughan et al., 2018). Furthermore, UHRF2 shows reduced preference for hemimethylated DNA (Pichler et al., 2011; Vaughan et al., 2018). These divergences may explain why the activity of UHRF2 is not general but directed at H3K9me3-marked regions. Additionally, UHRF2 has been identified as a 5hmC reader through its SRA domain (Spruijt et al., 2013; Zhou et al., 2014) and its H3 ubiquitin ligase activity is robustly stimulated by this mark (Vaughan et al., 2018). 5hmC is re-localized to repetitive elements during PGC development (Hill et al., 2018) and could also contribute to specifically direct UHRF2 activity at these sites. Lastly, the difference in selectivity could be due to unknown protein partners that target the action of UHRF2 in PGCs.”